# MoReFlow: Motion Retargeting Learning through Unsupervised Flow Matching

## Abstract

Motion retargeting holds a premise of offering a larger set of motion data for characters and robots with different morphologies. Many prior works have approached this problem via either handcrafted constraints or paired motion datasets, limiting their applicability to humanoid characters or narrow behaviors such as locomotion. Moreover, they often assume a fixed notion of retargeting, overlooking domain-specific objectives like style preservation in animation or task-space alignment in robotics. In this work, we propose **MoReFlow**, Motion Retargeting via Flow Matching, an unsupervised framework that learns correspondences between characters' motion embedding spaces. Our method consists of two stages. First, we train tokenized motion embeddings for each character using a VQ-VAE, yielding compact latent representations. Then, we employ flow matching with conditional coupling to align the latent spaces across characters, which simultaneously learns conditioned and unconditioned matching to achieve robust but flexible retargeting. Once trained, **MoReFlow** enables flexible and reversible retargeting without requiring paired data. Experiments demonstrate that **MoReFlow** produces high-quality motions across diverse characters and tasks, offering improved controllability, generalization, and motion realism compared to the baselines.

## 1 Introduction

Motion retargeting has been the fundamental problem of adapting motions performed by one actor to another with different morphology, such as transferring a human motion sequence to a virtual avatar or a humanoid robot. It holds the premise of an enriched dataset for virtual characters or robots by reducing the cost of content creation in animation and gaming, enabling robots to learn practical skills from human demonstrations, and supporting cross-actor motion analysis in biomechanics. Despite its diverse applications, the central challenge remains the same: establishing a reliable correspondence between the motion spaces of different characters while respecting different kinematic and dynamic capabilities.

Motion retargeting has been studied with diverse approaches, such as optimization methods with handcrafted constraints (Zakka, 2025; Choi & Ko, 2000), reinforcement learning (Reda et al., 2023), and learning-based mappings between characters (Villegas et al., 2021). However, most methods remain restricted to the same human morphologies (Aberman et al., 2020; Aigerman et al., 2022). While some recent works tackle motion retargeting between heterogeneous morphologies, such as human-to-quadruped (Li et al., 2024b;a), they are often limited to locomotion tasks. Moreover, existing approaches typically assume a fixed notion of retargeting. In practice, the objective of the retargeting may vary: motion retargeting in animation often means joint-space alignment to preserve style, while robotics applications may require task-space alignment to achieve functional goals. These limitations call for a more general and adaptable retargeting framework.

In this work, we propose **MoReFlow**, Motion Retargeting via Flow Matching, an unsupervised framework for learning motion retargeting across heterogeneous characters. Specifically, **MoReFlow** formulates the retargeting problem as constructing a correspondence between the motion spaces of two characters using Flow Matching. The framework consists of two stages: pretraining a motion tokenizer and learning motion correspondence via codebook flow. In the first stage, we train tokenized motion embeddings for each character using a VQ-VAE (Van Den Oord et al., 2017), which yields a compact motion representation together with a motion encoder and decoder. In the

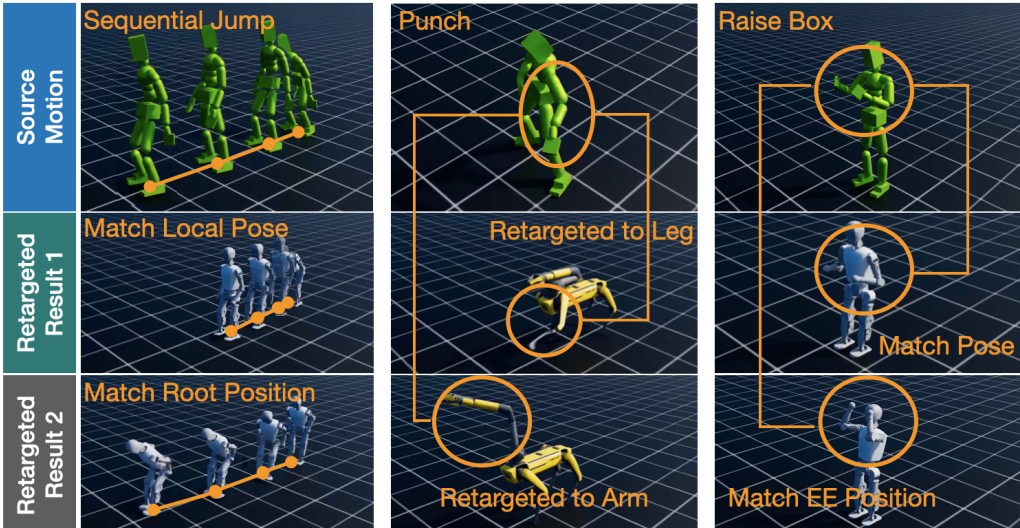

Figure 1: **MoReFlow** enables diverse source motions to be retargeted across different target characters with controllable outcomes. The source motions include both dynamic whole-body movements and fine-grained manipulation tasks. The target characters range from smaller humanoid robots to morphologically distinct platforms such as Spot. Our framework can generate multiple retargeted variations to accommodate different user preferences.

second stage, **MoReFlow** employs guided flow matching to establish correspondences between the latent spaces of different characters. At inference, the source character's motion is encoded into a latent vector, translated into the target character's latent vector through the learned flow, and then reconstructed by the target character's decoder. Our unsupervised framework employs conditional coupling to handle large unpaired datasets, where it guides the flow to minimize geodesic distances in the feature space between samples.

We conduct extensive experiments covering humanoid and non-humanoid robots to validate our framework. Our main results show that **MoReFlow** can retarget diverse dynamic human motions—including locomotion, sports, and gestural actions—to new characters such as the smaller Booster T1 humanoid and the Spot quadruped with a manipulator (Figure 1). The retargeted motions not only preserve the style and semantics of the source but also adapt naturally to the target morphology. Furthermore, our framework enables condition-dependent control, producing multiple valid outcomes from the same source motion depending on whether the alignment is defined in local or world coordinates in real time. Compared to the baseline methods, **MoReFlow** achieves better motion quality while offering a higher degree of controllability in cross-morphology retargeting. Finally, our ablation studies analyze the effects of various factors, such as different embedding designs and motion dataset volume.

In summary, our contributions are as follows:

- We introduce **MoReFlow**, a novel unsupervised framework for cross-morphology motion retargeting that leverages flow matching in tokenized motion spaces. It enables retargeting without paired data and supports reversible and modular mappings between characters.

- We show that the proposed framework can generate different motion retargeting results under various conditions, such as local style alignment or world-frame task alignment, which offer users interactive, fine-grained control.

- We validate the effectiveness of the proposed approach through extensive experiments. The results demonstrate that **MoReFlow** achieves controllable, high-quality motion retargeting that preserves semantic intent and outperforms baseline methods in both motion fidelity and controllability.

## 2 RELATED WORK

### 2.1 MOTION RETARGETING

Motion retargeting methods can be broadly divided into two categories: *homo-morphology* methods, where source and target share the same skeletal topology but differ in proportions or geometry, and *cross-morphology* methods, where source and target differ fundamentally in skeletal structure.

**Homomorphology Motion Retargeting.** Early learning-based approaches such as Neural Kinematic Networks (NKN) (Villegas et al., 2018) and Skeleton-Aware Networks (Aberman et al., 2020) pioneered unsupervised retargeting by incorporating kinematic layers and graph-structured features. Subsequent work extended these ideas to improve realism and robustness, for example through residual perception modules (Zhang et al., 2023b), contact- and geometry-aware constraints (Villegas et al., 2021), or correspondence-free streaming models for online transfer (Rekik et al., 2024). More recent studies leverage skeleton-agnostic embeddings and canonical templates to generalize across characters (Lee et al., 2023; Zhang et al., 2024), while generative approaches such as diffusion and collision-aware frameworks further enhance plausibility (Cao et al., 2025; Martinelli et al., 2024). In robotics, optimization-based methods such as MINK (Zakka, 2025) provide inverse-kinematics baselines for retargeting motions from human datasets to humanoid robots like NAO, Atlas, and Valkyrie.

**Cross-Morphology Motion Retargeting.** Retargeting across characters with different skeletal structures requires bridging more substantial morphological gaps. Adversarial correspondence embedding (ACE) (Li et al., 2023), phase-manifold alignment (Li et al., 2024a), and reinforcement learning frameworks such as CrossLoco (Li et al., 2024b) have been proposed to transfer human motions to quadrupeds and robots. Broader frameworks such as BuddyImitation (Li et al., 2025) extend retargeting to interaction skills by modeling relational dynamics. Other studies address human-to-quadruped transfer via imitation learning or hierarchical controllers (Kim et al., 2022; Yoon et al., 2024), while optimization-based approaches such as differentiable optimal control (DOC) exploit model predictive control for legged robots (Grandia et al., 2023). Unsupervised latent alignment methods (Yan et al., 2023; Mourot et al., 2023) and generative approaches (Cao et al., 2025) further demonstrate the diversity of strategies for handling structural mismatches in cross-embodiment motion retargeting.

### 2.2 FLOW MATCHING AND GENERATIVE FLOWS

Flow matching has recently emerged as a powerful alternative to diffusion models for generative modeling. Unlike diffusion, which relies on stochastic denoising processes, flow matching trains continuous normalizing flows by regressing vector fields along probability paths, offering both simulation-free training and efficient sampling (Lipman et al., 2022). The method has since been applied across a wide range of modalities, including image and video generation, audio synthesis, and text modeling (Lipman et al., 2024), with theoretical analyses highlighting its equivalence to Gaussian diffusion processes (Gao et al., 2024). More recently, flow matching has been adapted to motion domains. Hu et al. (2023) introduced Motion Flow Matching for efficient human motion generation and editing, while Cuba et al. (2025) extended the framework with conditional target-predictive flows to reduce temporal jitter in text-to-motion tasks. These works highlight the versatility of flow matching and motivate its use in our framework, where efficient and controllable latent-space alignment is central to motion retargeting.

Unlike GANs or VAEs, MoReFlow uses flow matching to ensure stable training without mode collapse or over-smoothing. Its continuous vector field uniquely enables reversible mapping and precise control between diverse retargeting conditions, offering a robust alternative to prior stochastic or adversarial methods.

## 3 MOTION RETARGETING VIA FLOW MATCHING

Our framework, **MoReFlow** addresses motion retargeting by learning a correspondence between the motion spaces of different characters in an unsupervised manner. An overview of the method is presented in Figure 2. The method consists of two main stages. In the first stage, we construct

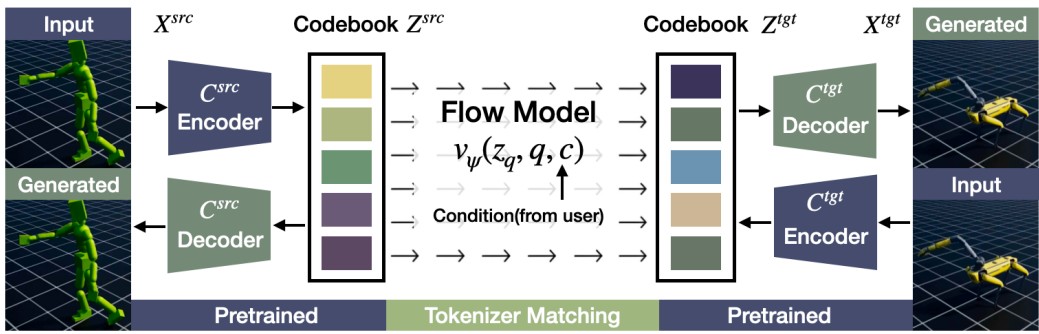

Figure 2: Overview of the proposed MoReFlow framework. Each character ($C^{src}$ and $C^{tgt}$) has a pretrained VQ-VAE tokenizer consisting of an encoder, a decoder, and a codebook. A source motion is first encoded and quantized into tokens from the source codebook. The flow matching model then maps the token distribution from the source codebook to the target codebook, optionally conditioned on task requirements, such as local style alignment or world-frame alignment. The retargeted motion is reconstructed using the pretrained target decoder.

compact motion embeddings for each character using a vector-quantized variational autoencoder (VQ-VAE). This tokenized representation provides a discrete latent space that captures motion patterns while discarding low-level redundancy. In the second stage, we employ flow matching to learn a transformation between the latent spaces of different characters. By combining these two stages, **MoReFlow** enables flexible motion retargeting across characters with different morphologies. Once trained, the framework supports diverse retargeting outcomes under different conditions, such as local style preservation or task-space alignment, and can also perform retargeting in reverse without requiring additional models.

### 3.1 PRETRAINED MOTION TOKENIZER

Directly performing motion retargeting in raw trajectory space is difficult because motion sequences are high-dimensional and redundant. To simplify the problem, we construct a compact tokenized representation of motion using a VQ-VAE inspired by T2M-GPT (Zhang et al., 2023a). Each input motion frame is represented as $x_t = [p_t; r_t; v_t^{\text{root}}]$ where $p_t$ and $r_t$ denote joint positions and rotations, and $v_t^{\text{root}}$ denotes the root velocity at frame $t$. Instead of encoding single frames, we partition the sequence into overlapping motion chunks of length $H$. Each chunk $x_{t:t+H-1} = \{x_t, \ldots, x_{t+H-1}\}$ is processed by an encoder $E_\theta$ that produces a latent feature

$$z_t = E_\theta(x_{t:t+H-1}) \in \mathbb{R}^d. \tag{1}$$

The latent vector is quantized using a codebook $\mathcal{E} = \{e_1, \ldots, e_K\}$ of $K$ embeddings. Each $z_t$ is replaced by its nearest entry and the decoder $D_\phi$ reconstructs the motion chunk from quantized tokens:

$$\hat{z}_t = \arg\min_{e_k \in \mathcal{E}} \|z_t - e_k\|_2, \tag{2}$$

$$\hat{x}_{t:t+H-1} = D_\phi(\hat{z}_t). \tag{3}$$

The model is trained with the standard VQ-VAE objective that consists of three terms: reconstruction loss $\mathcal{L}_{\text{rec}}$, codebook loss $\mathcal{L}_{\text{code}}$ and commitment loss $\mathcal{L}_{\text{commit}}$:

$$\mathcal{L}_{\text{VQ}} = \mathcal{L}_{\text{rec}} + \mathcal{L}_{\text{code}} + \mathcal{L}_{\text{commit}}, \tag{4}$$

$$\mathcal{L}_{\text{rec}} = \|x_{t:t+H-1} - \hat{x}_{t:t+H-1}\|^2, \tag{5}$$

$$\mathcal{L}_{\text{code}} = \|\text{sg}[z_t] - e_k\|^2, \tag{6}$$

$$\mathcal{L}_{\text{commit}} = \beta\|z_t - \text{sg}[e_k]\|^2 \tag{7}$$

where $\text{sg}[\cdot]$ denotes the stop-gradient operator and $\beta$ is a weighting coefficient. This process yields a sequence of motion tokens that capture essential spatiotemporal patterns while discarding redundancy. This discrete latent representation provides a compact and structured space that simplifies

learning correspondences across characters via flow matching. For each character, we train an individual set of an encoder, a decoder and a codebook.

We opt for this decoupled design because enforcing a shared latent space across heterogeneous morphologies (e.g., bipeds and quadrupeds) creates a bottleneck that often leads to *representation collapse*, where a single model fails to approximate distinct kinematic manifolds. By learning independent, morphology-specific codebooks, we ensure high-fidelity reconstruction for each embodiment while enabling modular scalability; adding a new robot simply requires training a lightweight tokenizer without retraining the entire framework.

## 3.2 MOTION RETARGETING VIA CODEBOOK FLOW

Once motion embeddings are obtained, we perform retargeting entirely in the latent space of motion tokens. Given a source motion $x^{\text{src}}_{t:t+H-1}$, the pretrained encoder first maps it into a latent representation $z^{\text{src}} = E^{\text{src}}_\theta(x^{\text{src}}_{t:t+H-1})$. Each latent vector is then quantized by selecting its nearest codebook entry from the source character's codebook $\mathcal{E}^{\text{src}}$ as defined in Equation 2.

We treat each quantized motion token $\hat{z}^{\text{src}}$ as a categorical distribution over source tokens. The retargeting model $f_\psi$ is conditioned on a variable $c$ that acts as an alignment objective selector (e.g., "Local Pose" vs. "World Task"). Crucially, $c$ serves a dual role: during training, it selects the specific feature function $\Phi(\cdot, c)$ used to compute the coupling cost and loss; during inference, it guides the flow field to generate motions satisfying that specific geometric constraint. It defines a time-dependent velocity field $v_\psi(\cdot, q, c)$ that transports the distribution to the target token space. At inference time, the process starts from the source distribution $p^{\text{src}}$ at $q = 0$ and integrates the following ODE until $q = 1$:

$$\frac{d}{dt} z_q = v_\psi(z_q, q, c), \quad z_0 = z^{\text{src}}. \tag{8}$$

Numerical integration produces the final distribution $p_1 \approx p^{\text{tgt}}$. From this, the most likely token sequence $\tilde{z}^{\text{tgt}}$ is extracted and decoded by the pretrained target decoder to yield the retargeted motion:

$$\hat{x}^{\text{tgt}} = D^{\text{tgt}}_\phi(\tilde{z}^{\text{tgt}}). \tag{9}$$

Condition-dependent motion retargeting is crucial in practice. For example, when aligning motion style in animation, the condition $c$ may specify local-frame style alignment so that the target reproduces the same movement pattern regardless of global trajectory. In contrast, motion retargeting for robotic manipulation or navigation tasks may require condition to specify world-frame alignment to ensure that end-effectors or base trajectories match global objectives. To support multi-conditioned motion retargeting, **MoReFlow** employs classifier-free guidance (Ho & Salimans, 2022) to strengthen the influence of conditions without requiring explicit external classifiers. During inference, the velocity field is evaluated both with and without the condition $c$, and the two predictions are interpolated:

$$v^{\text{guided}}_\psi(z_q, q, c) = (1 - \gamma) \, v_\psi(z_q, q, \varnothing) + \gamma \, v_\psi(z_q, q, c), \tag{10}$$

where $\gamma$ is the guidance scale. This strategy allows users to smoothly control the strength of condition signals, enabling flexible trade-offs between different motion retargeting preferences.

## 3.3 LEARNING MOTION RETARGETING VIA UNSUPERVISED FLOW MATCHING

Training the retargeting model $f_\psi$ is challenging because we do not assume access to paired motion datasets across characters. To address this, we combine flow matching with condition-dependent feature regularization.

We consider the distributions $z^{\text{src}}$ and $z^{\text{tgt}}$ over the source and target codebooks. Flow matching defines an interpolating distribution $p_t$ between $z^{\text{src}}$ and $z^{\text{tgt}}$ by setting the velocity $\dot{z}$ as $z^{\text{tgt}} - z^{\text{src}}$, and the model learns a velocity field $v_\psi$ to match the probability flow Lipman et al. (2024):

$$\mathcal{L}_{\text{FM}} = \mathbb{E}_{t \sim \mathcal{U}[0,1]} \|v_\psi(z_q, q, c) - \dot{z}_q\|^2. \tag{11}$$

To guide alignment toward meaningful retargeting outcomes, we define a feature extractor $\Phi(\cdot, c)$ that computes condition-specific descriptors from a motion sequence, where the details of the feature

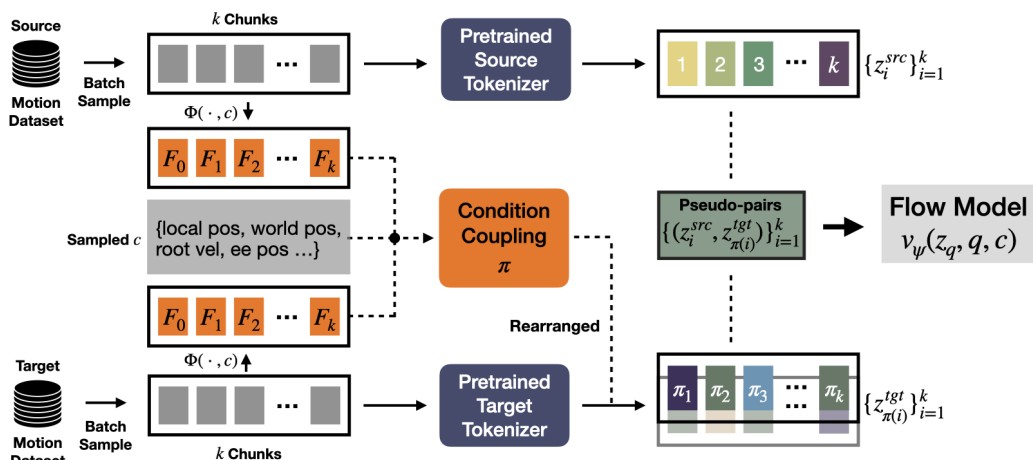

Figure 3: Overview of the Multi-Sample Condition Coupling process A.3 in every training step

functions are provided in A.4. The feature loss compares the extracted features of the source motion $x^{\text{src}}$ and the retargeted motion $\hat{x}^{\text{tgt}}$:

$$\mathcal{L}_{\text{feat}} = \|\Phi(x^{\text{src}}, c) - \Phi(\hat{x}^{\text{tgt}}, c)\|^2. \tag{12}$$

This formulation remains flexible across diverse conditioning schemes without committing to fixed feature types.

Following the classifier-free guidance paradigm, during training we randomly drop the condition $c$ (with probability $p_{\text{mask}}$), replacing it with a null condition $\varnothing$. This allows the model to learn both unconditional and conditional velocity fields, enabling stronger control during inference via interpolation (Zheng et al., 2023). The final objective combines the flow matching loss and the condition-dependent feature loss with a weighting variable $\alpha$:

$$\mathcal{L}_{\text{total}} = \mathcal{L}_{\text{FM}} + \alpha \mathcal{L}_{\text{feat}}. \tag{13}$$

To enable unsupervised learning without paired data, we introduce **Multi-Sample Condition Coupling**, as illustrated in Figure 3. Instead of relying on fixed correspondences, we sample independent batches and compute a pairwise cost matrix using the condition-specific feature metric $d_{ij} = \|\Phi(x_i^{src}, c) - \Phi(x_j^{tgt}, c)\|^2$, solving for an optimal assignment to form "pseudo-pairs." This dynamic coupling acts as a stochastic regularization mechanism: by associating the same source motion with different, semantically similar target motions across training steps, it provides an effective "one-to-many" data augmentation. This forces the flow model to learn generalized semantic directions rather than memorizing specific trajectory pairs, thereby preventing mode collapse and ensuring robust generalization to unseen data. The details of multi-sample condition coupling and pseudo-pairs are provided in A.3.

## 4 EXPERIMENT

In this section, we evaluate **MoReFlow** through a comprehensive set of experiments. We begin by presenting our main results, which demonstrate that MoReFlow can retarget diverse dynamic human motions to both humanoid and quadrupedal robots with high motion quality, semantic alignment, and condition-dependent controllability. Next, we quantitatively compare our approach against several baseline methods, showing that **MoReFlow** achieves superior performance in cross-morphology motion retargeting while offering greater flexibility. Finally, we conduct a series of ablation studies to analyze the impact of key design choices such as embedding structure, motion dataset volume.

### 4.1 EXPERIMENT SETUP

We train and evaluate on the AMASS dataset (Mahmood et al., 2019), using a total of 482 clips corresponding to about 2,747 seconds or 82,410 frames. For **locomotion/lower-body** motions, we use 117 human source clips and 258 target-domain clips to cover a broad range of speeds and headings.

For **arm/upper-body** motions, we use 105 human source clips and 310 target-domain clips to span diverse end-effector paths. The characters for source and target include an SMPL-humanoid (human), a half-sized SMPL-humanoid (half-human), the Booster T1 robot (T1) (Booster Robotics), and the Spot quadruped (Spot) (Boston Dynamics, 2019). The detail of dataset distribution is in Appendix A.5

## 4.2 Implementation Details

**Tokenization.** We train a separate motion VQ-VAE tokenizer for each character, which converts 32-frame motion windows (=1 second) into compact token sequences. After normalizing each clip using the dataset-level mean and standard deviation, we extract all valid 32-frame windows for training and evaluation. VQ-VAE consists of a temporal convolutional encoder–decoder with configurable depth and width, which is combined with an EMA-reset quantizer (Williams et al., 2020). For SMPL humanoids and the Booster T1 robot, we use a codebook size of $512 \times 512$, while for the Spot quadruped we use $256 \times 256$. Temporal downsampling by a factor of 4 reduces each 32-frame window to a short latent sequence of 8 tokens, which serves as the basis for retargeting. Codebook sequences are generated in batches for efficiency. For training, we use the AdamW optimizer (Loshchilov & Hutter, 2017) with betas [0.9, 0.99], a batch size of 128, and an exponential moving average constant $\mu = 0.99$. The Motion VQ-VAEs are trained with a learning-rate warm-up of 1K iterations at 2e-4, followed by 100K iterations at 1e-5.

**Flow Matching.** Our Discrete-Flow-Transformer maps source tokens to target tokens conditioned on task features. It employs token embeddings, sinusoidal positional encoding, and separate time/condition MLP embeddings, followed by a Transformer encoder (Vaswani et al., 2017). The encoder is configured with 512-dimensional token embeddings, 6 encoder layers $\times$ 8 self-attention heads, a 2048-dimensional feedforward sublayer, a dropout rate of 0.1, and a maximum sequence length of 8 tokens. The encoder output is connected to a linear prediction head over the target vocabulary. We use the same AdamW optimizer with a learning rate of 1e-4 and betas [0.9, 0.99] for a total 200K iterations with a 1K warm-up using the cosine LR scheduler (Loshchilov & Hutter, 2016), and the weight decay is 1e-2. The coupling batch size is set to $k = 512$.

All experiments run on a single RTX 4090 with an Intel i9-13900K (24 cores). Training the motion VQ-VAEs takes less than 1 hour per character, and training the discrete flow retargeter with 5 different conditions A.4 takes 8 hours approximately.

## 4.3 Main Results

We first present the main results of **MoReFlow**. Our framework successfully retargets a wide range of dynamic human motions, including locomotion, jumping, tennis and golf swings, and boxing, onto the half human, T1, and Spot robots. For example, human walking motions following ICLR-shaped trajectories are retargeted to all three characters, producing smooth and physically consistent executions. In addition, **MoReFlow** transfers challenging upper-body and whole-body motions with physical realism, demonstrating its ability to handle a broad spectrum of dynamic motions. For the details, please refer to the supplementary video.

A key advantage of **MoReFlow** is its controllability. Unlike traditional methods that typically produce a fixed retargeted motion for a given condition, our framework allows users to specify alignment conditions and thereby generate different outcomes. For example, the same human walking motion can be retargeted onto a smaller half-human or T1 in different ways, such as preserving the original style or aligning the global trajectory through faster walking or running (Figure 1, 1st column). In fact, task-space alignment can also induce drastically different behaviors, such as T1 jumping to match human height, T1 jumping more dynamically to follow a human leap, or a human crawling to match the lower height of Spot. For upper-body gestures, **MoReFlow** allows users to toggle the mapping of left, right, or both arms for T1 waving, or to direct the mapping to either a manipulator or a front leg for Spot boxing (Figure 1, 2nd column). Even in a book-retrieval scenario involving full-body motions, T1 can either stylistically imitate the human in the local frame or match task-space alignment by adjusting shoulder angles and reaching a higher shelf (Figure 1, 3rd column). Such controllability makes **MoReFlow** broadly applicable across domains where users may prioritize either stylistic fidelity or task effectiveness.

Another advantage of **MoReFlow** lies in its ability to support chain-of-retargeting, which enables motion transfer between characters that do not share a direct mapping. Because the flow formulation is reversible, motions can be transferred modularly through intermediate characters. For example, we demonstrate the transfer of a quadruped robot's hand-waving motion to the Booster T1 robot's left and right hands by first retargeting the motion back to the human domain, and then mapping it onto the new robot.

### 4.4 BASELINE COMPARISON

To further validate the effectiveness of **MoReFlow**, we compare it against several baseline methods. The evaluation includes both qualitative visualizations and quantitative metrics that measure motion quality and controllability.

To select baselines, we focus on cross-embodiment motion retargeting algorithms in an unsupervised setting. We exclude other approaches as they typically rely on homologous morphologies, paired data, or task-specific RL rewards. Thus, **Walk-the-Dog (WtD)** (Li et al., 2024a) and **ACE** (Li et al., 2023) represent the recent state-of-the-art baselines that share our challenging setting of learning unsupervised cross-morphology retargeting via motion priors. **WtD** introduces a vector-quantized periodic autoencoder to learn a shared phase manifold across different characters, enabling alignment in both timing and semantics. It also learns a shared latent codebook for both source and target characters to cluster semantically similar motions. In contrast, our **MoReFlow** framework trains separate codebooks for each character and establishes correspondences between them through a learned flow field, which provides greater flexibility and modularity. On the other hand, **ACE** adopts a two-stage paradigm similar to ours: it first trains a motion prior, then learns a retargeting networks. However, **ACE** learns the retargeting function in a single step using a GAN-based adversarial objective. It also builds the motion prior using a motion VAE, whereas **MoReFlow** employs a VQ-VAE to obtain a tokenized latent representation. For all baselines, we follow the implementations provided in the original papers, with minor adaptations to ensure compatibility with our experimental setup.

For quantitative evaluation, we employ the following metrics:

- **FID**: computes the Fréchet Inception Distance between the feature distributions of source and retargeted motions, assessing overall distributional similarity. Lower FID values signify that the retargeted motion more closely resemble the source motion.

- **Diversity (DIV)**: measures the variability of retargeted motions across different sequences, capturing the expressiveness of the model.

- **Alignment (ALI)**: measures feature-level consistency between the source and retargeted motions by computing differences in extracted features.

- **Naturalness (NAT)**: assesses whether the generated motions are physically plausible. We identify unnatural frames, such as those exhibiting foot sliding, foot penetration, or joint vibration, and report the percentage of natural frames as the score.

In summary, *FID* and *NAT* primarily evaluate motion quality in terms of realism and physical plausibility, whereas *DIV* and *ALI* predominantly assess controllability, capturing expressive diversity and semantic consistency under fixed conditions.

A summary of the results is presented in Table 1. Overall, our results show that **MoReFlow** consistently outperforms the baseline methods in both motion quality and controllability. Compared to **WtD**, our approach does not rely on an internal periodic structure to guide motion generation. While **WtD**'s design enforces smooth periodic behaviors, it limits the ability to encode the full diversity of character motions. **WtD** performs well on periodic locomotion by aligning timing with a shared phase–codebook and frequency scaling, but it degrades markedly on non-periodic upper-body motions. For example, in lifting a box, **WtD** under-represents the amplitude and timing of elbow extension–flexion, leading to insufficient reach and frequent over/undershoot during the hold phase. In contrast, our method leverages character-specific codebooks connected through flow matching, which allows for more expressive movements. This is evident in our results, where **MoReFlow** achieves higher *Diversity* and stronger *Alignment* scores than **WtD**.

| Method | FID ↓ | | DIV ↑ | | ALI ↑ | | NAT (%) ↑ | |
|---|---|---|---|---|---|---|---|---|
| | Loco. | Upper. | Loco. | Upper. | Loco. | Upper. | Loco. | Upper. |
| WtD-human | 42.3 | 77.1 | 0.44 | 0.39 | 0.82 | 0.57 | 86.0 | 81.8 |
| ACE-human | 50.4 | 61.1 | 0.54 | 0.52 | 0.77 | 0.73 | 82.4 | 84.2 |
| **MoReFlow**-human | **32.4** | **35.8** | **0.69** | **0.64** | **0.87** | **0.80** | **89.9** | **93.5** |
| WtD-non-human | 42.5 | 72.9 | 0.46 | 0.41 | **0.85** | 0.54 | 88.4 | 82.6 |
| ACE-non-human | 53.0 | 60.6 | 0.58 | 0.50 | 0.80 | 0.77 | 79.3 | 84.0 |
| **MoReFlow**-non-human | **33.8** | **42.0** | **0.68** | **0.62** | 0.83 | **0.79** | **89.2** | **92.3** |

Table 1: Comparison of MoReFlow against baseline methods for Human and Non-human targets on Locomotion (Loco.) and Upper-body retargeting tasks.

When compared to **ACE**, **MoReFlow** also demonstrates clear advantages. **ACE** constructs its motion prior using a MotionVAE, which can suffer from posterior collapse, causing the generated motion to depend primarily on the character's previous pose rather than the source motion. In addition, **ACE** performs retargeting as a single-step GAN-based mapping, which often produces unstable results. By contrast, **MoReFlow** generates retargeted motions gradually through the flow matching process, yielding smoother transitions and greater stability. With world-frame alignment on a half-sized SMPL-humanoid, **ACE**'s prior-state dependency delays step-length and velocity adaptation, leading to increased toe slip, whereas **MoReFlow** promptly adjusts these parameters via condition-guided vector fields, reducing slip. On Spot with a manipulator for tennis swinging, **ACE**'s single-step discriminator causes wrist jitter and impact undershoot, while **MoReFlow** integrated flow through peak velocity, deceleration, and stop improves impact stability. This design directly contributes to **MoReFlow** achieving lower *FID* and higher *NAT* scores than **ACE**, as the generated motions better preserve semantic intent while avoiding artifacts such as foot sliding or joint vibration.

For locomotion, **WtD**'s phase manifold and frequency-scaled matching perform well in capturing periodic timing and semantics (*ALI*), but its shallow decoder and shared codebook limit distributional *FID* and *DIV* scores.Specifically, **WtD** achieves a slightly higher alignment score in non-human locomotion, as its explicit phase modeling is particularly effective for rhythmic quadruped gaits. For upper-body tasks, which are less periodic, both **WtD** and **ACE** exhibit reduced naturalness (*NAT*)—**WtD** due to phase bias and **ACE** due to adversarial instability. In contrast, **MoReFlow**'s character-specific codebooks and continuous flows preserve semantics while mitigating artifacts, resulting in lower *FID* and higher *NAT*.

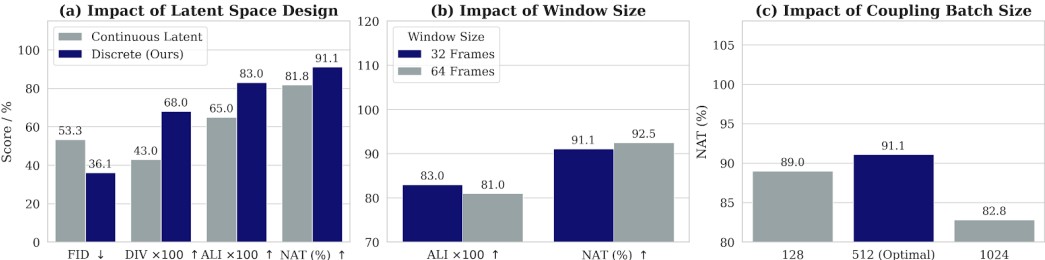

Figure 4: Ablation studies on key design choices. (a) Latent Space: Our discrete tokenization significantly outperforms continuous latents in both generation quality (FID) and controllability (DIV, ALI). (b) Window Size: A 32-frame window offers the optimal trade-off between semantic alignment and motion naturalness. (c) Coupling Batch Size: A batch size of 512 yields the highest motion realism.

## 4.5 ABLATION STUDY

To further understand the design choices in **MoReFlow**, we conduct a series of ablation studies. These experiments examine how the latent representation, window size (chunk size), and coupling batch size affect retargeting quality and controllability. By comparing different configurations, we provide insight into why our final configuration achieves both high-quality and flexible motion retargeting.

**Latent space design.** We first analyze the role of the latent space in motion retargeting. In this experiment, we compare three configurations: (1) retargeting directly in trajectory space without using a latent embedding, (2) retargeting in a continuous latent space without quantization, and (3) our proposed design using a VQ-VAE tokenized latent space. The results show that retargeting without a latent space suffers from unstable training and poor motion quality, as the high dimensionality of trajectories makes alignment difficult. The continuous latent variant improves stability, which result *FID* value lower (53.3) and *NAT* higher as **ACE**, 81.8% in Figure 4 (a). But still produces motions with degraded diversity and alignment. In contrast, our discrete tokenized latent space provides compact and expressive motion representations, yielding the best performance across FID, Diversity, and Alignment metrics.

**Window size.** The window size controls the temporal context seen by both the tokenizer and our Discrete-Flow-Transformer. Smaller windows emphasize local kinematics but may truncate non-periodic transients; larger windows capture longer dependencies, but increase token-sequence length and matching ambiguity unless the tokenizer's temporal downsampling and the Discrete-Flow-Transformer's maximum sequence length are co-tuned. Empirically, 32 frames worked best with nearest-neighbor (Euclidean) coupling driven by root-velocity features (local, short-horizon signals), while 64 frames benefited optimal transport (cosine) coupling in pattern space (longer temporal templates). With nearest-neighbor coupling, 32 frames yields the best semantic controllability with competitive realism, yields the highest *DIV, ALI* values as Table 1, whereas with optimal transport, 64 frames improves realism (*NAT* 92.5%) and stability at a slight diversity cost (*ALI* goes down to 0.81) in Figure 4 (b). These settings balanced expressivity and stable couplings without exceeding the flow matching sequence budget. See Appendix A.3 for the per-step coupling procedure.

**Coupling batch size.** Coupling is computed within each mini-batch: increasing the number of samples per step yields a denser bipartite candidate set, improving one-to-one pairing quality and reducing variance of the training signal; however, both cost and noise sensitivity grow with batch size, especially for transport-based coupling. We found a clear "sweet spot": 512 for nearest-neighbor coupling, which offers sufficient candidate coverage without oversmoothing, and 256 for optimal transport coupling, which captures adequate structure while avoiding heavy, noisy transport in very large batches Figure 4 (c). Beyond these points, we observed diminishing returns and sporadic instability. When coupling batch size grows to the sweet spot, we observe diminishing returns and mild degradation in *NAT* with higher training variance.

# 5 LIMITATIONS

A key limitation of our current framework is that motion retargeting is performed in a one-to-one manner, where a flow model is trained specifically for each source–target character pair. This design simplifies the learning problem and allows us to achieve high-quality results, but it prevents the framework from directly generalizing to many-to-many retargeting. Although retargeting between arbitrary characters can in principle be achieved by chaining intermediate mappings, this strategy introduces inefficiency: the computational cost grows with the number of intermediate characters, and accumulated errors along the chain may degrade motion fidelity. Moreover, such chaining assumes that a suitable path of characters exists between the source and target, which is not guaranteed in practice. Extending **MoReFlow** toward a unified framework that enables flexible one-to-many or any-to-any retargeting, while maintaining scalability and motion quality, is an important direction for future work.

# 6 CONCLUSION

In this work, we introduced MoReFlow, an unsupervised framework for cross-morphology motion retargeting based on flow matching in tokenized motion spaces. By combining character-specific VQ-VAE embeddings with condition-dependent flow models, our approach enables controllable and reversible motion transfer across a wide range of characters without requiring paired data. Extensive experiments demonstrate that MoReFlow not only achieves high-quality retargeting results across humanoid and quadrupedal robots but also offers explicit controllability to adapt motions to different alignment requirements. Compared to prior baselines, our framework delivers superior motion fidelity, diversity, and naturalness while providing deeper engineering insights through ablation studies. We believe MoReFlow establishes a scalable foundation for generalizable cross-character motion retargeting, and future work will explore extending the framework to many-to-many retargeting and real-world robot deployment.

## REPRODUCIBILITY STATEMENT

We have made extensive efforts to enhance the reproducibility of our work, including providing hyperparameters, detailed formulations of conditions and pseudocode for conditional coupling in the Appendix. In addition, the code will be made publicly available upon acceptance.

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

# A TRAINING HYPERPARAMETERS

## A.1 VQ-VAE (TOKENIZER) PER CHARACTER

Table 2

| Group | Param | Value |
|---|---|---|
| Data | window_size | 32 |
| | batch_size | 128 |
| | num_workers | 4 |
| Optimization | total_iter | 100,000 |
| | warmup_iter | 1,000 |
| | lr | 2e-4 |
| | optimizer | AdamW |
| | betas | (0.9, 0.99) |
| | weight_decay | 0.0 |
| | lr_scheduler | [50k, 100k] |
| | scheduler $\gamma$ | 0.05 |
| Loss | recons_loss | l1_smooth |
| | commit $\beta$ | 0.02 |
| Quantizer | quantizer | ema_reset |
| | $\mu$ | 0.99 |
| | beta | 1.0 |
| Humanoids (SMPL, half, Booster T1) | nb_code | 512 |
| | code_dim | 512 |
| | width/depth | 512 / 3 |
| | down_t / stride_t | 2 / 2 |
| | out_emb_width | 512 |
| Spot quadruped | nb_code | 256 |
| | code_dim | 256 |
| | width/depth | 256 / 3 |
| | down_t / stride_t | 2 / 2 |
| | out_emb_width | 256 |
| Misc | activation / norm | relu / none |
| | seed | 123 |

Table 2: VQ-VAE tokenizer hyperparameters per character family.

## A.2 FLOW MATCHING RETARGETER

Table 3

| Group | Param | Value |
|---|---|---|
| Discrete tokens | `vocab_size` | = *target* `nb_code` |
| | `max_seq_len` | 8 |
| Transformer | `d_model` | 512 |
| | `n_layers` | 6 |
| | `n_head` | 8 |
| | `d_ff` | 2048 |
| | `dropout` | 0.1 |
| | `activation` | GELU |
| | `pos. encoding` | sinusoidal |
| | `batch_first` | `True` |
| Cond/Time embed | `time_embed` | 2-layer MLP, SiLU |
| | `cond_embed` | 2-layer MLP, SiLU |
| | `cond_dim` | derived |
| Data | `window_size` | 32 (default) |
| | `coupling_batch_size` | 512 (NN) / 256 (OT) |
| | `num_workers` | 4 |
| Optimization | `total_iter` | 200,000 |
| | `lr` | 1e-4 |
| | `scheduler` | `CosineAnnealingLR` |
| Loss | $\mathcal{L}_{\text{FM}}$ | 1.0 |
| | $\mathcal{L}_{\text{feat}}$ | 0.2 |

Table 3: Flow Matching hyperparameters and Transformer configuration.

## A.3 MULTI-SAMPLE CONDITION COUPLING

---

**Algorithm 1** Pseudocode of Multi-Sample Condition Coupling

---

**Require:** Pretrained source tokenizer $E^{src}$, Pretrained target tokenizer $E^{tgt}$
**Require:** Source motion dataset $\mathcal{D}_{src}$, Target motion dataset $\mathcal{D}_{tgt}$
**Require:** Set of conditions $\mathcal{C}$ (including null condition $\emptyset$)
**Require:** Condition-specific feature extractor $\Phi(\cdot, c)$
 1: **Initialize:** Flow model parameters $\psi$
 2: **for** each training iteration **do**
 3:     // 1. Sample a condition and motion batches
 4:     Sample a condition $c \sim \mathcal{C}$
 5:     Sample a source motion batch $\mathcal{B}_{src} = \{x_i^{src}\}_{i=1}^k \sim \mathcal{D}_{src}$
 6:     Sample a target motion batch $\mathcal{B}_{tgt} = \{x_j^{tgt}\}_{j=1}^k \sim \mathcal{D}_{tgt}$
 7:
 8:     // 2. Compute condition-specific features for coupling
 9:     Compute source features $\{F_i^{src} \leftarrow \Phi(x_i^{src}, c)\}_{i=1}^k$
10:     Compute target features $\{F_j^{tgt} \leftarrow \Phi(x_j^{tgt}, c)\}_{j=1}^k$
11:
12:     // 3. Find correspondences by computing pairwise couplings
13:     Compute a cost matrix $M$ where $M_{ij} = \text{distance}(F_i^{src}, F_j^{tgt})$
14:     Find an optimal permutation $\pi = \text{argmin}_\sigma \sum_{i=1}^k M_{i,\sigma(i)}$
15:
16:     // 4. Form pseudo-pairs and update the flow model
17:     Create pseudo-pairs $\mathcal{P} = \{(x_i^{src}, x_{\pi(i)}^{tgt})\}_{i=1}^k$
18:     Obtain latent codes for pairs: $z_i^{src} \leftarrow E^{src}(x_i^{src}), z_{\pi(i)}^{tgt} \leftarrow E^{tgt}(x_{\pi(i)}^{tgt})$
19:     Compute loss $\mathcal{L}_{total}$ using $\{(z_i^{src}, z_{\pi(i)}^{tgt})\}_{i=1}^k$ and condition $c$ (from Eq. 13)
20:     Update parameters: $\psi \leftarrow \text{update}(\psi, \nabla_\psi \mathcal{L}_{total})$

---

**Detailed Description of Steps.** We provide a detailed breakdown of the multi-sample condition coupling process outlined in Algorithm 1.

**Step 1: Independent Sampling.** Unlike paired training, we sample the source batch $\mathcal{B}_{src}$ and target batch $\mathcal{B}_{tgt}$ independently from their respective datasets. Each batch consists of $k$ motion windows, where each window $x$ has a temporal length of $T = 32$ frames. This independence allows the model to explore "one-to-many" relationships, as a source motion sampled in epoch $t$ may be paired with a different target motion in epoch $t + 1$.

**Step 2: Feature Extraction.** To measure semantic similarity, we map the raw motion windows into a condition-dependent feature space. Given a condition $c$ (e.g., "Root Velocity"), the extractor $\Phi(\cdot, c)$ computes a feature vector $F \in \mathbb{R}^{d_{feat}}$ that summarizes the relevant kinematic properties (e.g., average velocity, end-effector positions) for that specific window, discarding irrelevant variations.

**Step 3: Pairwise Coupling.** This step establishes the correspondence between the independent batches. We first compute a pairwise cost matrix $M \in \mathbb{R}^{k \times k}$ representing the dissimilarity between every source-target pair in the batch:

$$M_{ij} = ||F_i^{src} - F_j^{tgt}||_2^2 \tag{14}$$

Based on this cost matrix, we determine the assignment $\pi$ using one of two strategies:

- **Nearest Neighbor (NN) Coupling:** This method greedily assigns the closest target sample to each source sample. It is computationally efficient and effective for local alignment but does not guarantee a one-to-one mapping (i.e., multiple source items may map to the same target).

$$\pi(i) = \underset{j \in \{1, \ldots, k\}}{\operatorname{argmin}} M_{ij} \tag{15}$$

- **Optimal Transport (OT) Coupling:** This method solves the linear sum assignment problem (LSAP) to find a global optimal permutation that minimizes the total batch cost. This enforces a strictly one-to-one bijective mapping, ensuring that the distribution of source motions is matched to the distribution of target motions within the batch. We solve this using the Hungarian algorithm:

$$\pi = \underset{\sigma \in S_k}{\operatorname{argmin}} \sum_{i=1}^{k} M_{i, \sigma(i)} \tag{16}$$

where $S_k$ is the set of all permutations of $\{1, \ldots, k\}$.

**Step 4: Update Flow Model.** Once the "pseudo-pairs" $\{(x_i^{src}, x_{\pi(i)}^{tgt})\}$ are formed, we treat them as ground-truth pairs for the current iteration. We encode them into discrete latent codes using the pre-trained VQ-VAE tokenizers ($z^{src} = E^{src}(x^{src})$, $z^{tgt} = E^{tgt}(x^{tgt})$) and update the flow matching model to minimize the discrepancy between the transported source distribution and the target distribution.

A.4 DETAILS OF CONDITIONS

**Notation.** Let $\Delta t = 1/\text{FPS}$, $R_t \in SO(3)$ be the world-frame root orientation at time $t$, and $p_t^{\text{root}} \in \mathbb{R}^3$ the world-frame root (pelvis) position. For any world-frame vector $x_t^{\text{world}}$, the root-aligned local coordinate is

$$x_t^{\text{local}} = R_t^\top x_t^{\text{world}}.$$

Let $p_t(j) \in \mathbb{R}^3$ denote the world-frame position of joint/end-effector $j$. Given a window of length $T$ (e.g., $T{=}32$), define the window average as

$$\Phi_{t:T}^{(\cdot)} = \frac{1}{T} \sum_{u=t}^{t+T-1} (\cdot).$$

**1. Root velocity condition (world-frame measurements)** Combine root linear and angular velocities expressed in the *root-aligned* frame (values retain physical units).

$$\mathbf{v}_t = \frac{R_t^\top\big(p_{t+1}^{\text{root}} - p_t^{\text{root}}\big)}{\Delta t} \in \mathbb{R}^3,$$

$$\boldsymbol{\omega}_t = \frac{\text{vee}\big(\log(R_t^\top R_{t+1})\big)}{\Delta t} \in \mathbb{R}^3,$$

$$\mathbf{c}_t^{\text{root}} = \big[\,\mathbf{v}_t \ ; \ \boldsymbol{\omega}_t\,\big] \in \mathbb{R}^6,$$

$$\Phi_{t:T}^{\text{root}} = \frac{1}{T}\sum_{u=t}^{t+T-1}\mathbf{c}_u^{\text{root}}.$$

*2D planar variant (optional).* If needed for planar walking:

$$\mathbf{c}_t^{\text{root-2D}} = \Big[\,(\mathbf{v}_t)_{x,y} \ ; \ \dot{\psi}_t\,\Big],$$

$$\dot{\psi}_t = \frac{\text{wrap}(\psi_{t+1} - \psi_t)}{\Delta t}.$$

**2. Local end-effector position condition (only limb-length normalization)** Use *root-aligned relative* EE positions. For an EE set $J$, each with an anchor $a_j$ (e.g., shoulder for wrist/elbow), define

$$\mathbf{r}_t(j \mid a_j) = R_t^\top\big(p_t(j) - p_t(a_j)\big) \in \mathbb{R}^3,$$

$$\widehat{\mathbf{r}}_t(j \mid a_j) = \frac{\mathbf{r}_t(j \mid a_j)}{\ell_{a_j \to j} + \varepsilon},$$

where $\ell_{a_j \to j} > 0$ is the limb length (measured at rest or averaged), and $\varepsilon$ is a small constant for numerical stability.

$$\mathbf{c}_t^{\text{local}} = \bigoplus_{j \in J}\widehat{\mathbf{r}}_t(j \mid a_j),$$

$$\Phi_{t:T}^{\text{local}} = \frac{1}{T}\sum_{u=t}^{t+T-1}\mathbf{c}_u^{\text{local}}.$$

*Arm-specific example (anchor = shoulder).*

$$\mathbf{c}_t^{\text{arm-local}} = \Big[\,\widehat{\mathbf{r}}_t(\text{wrist} \mid \text{shoulder}) \ ; \ \widehat{\mathbf{r}}_t(\text{elbow} \mid \text{shoulder})\,\Big].$$

When we want to make different conditions for each part of character body, $c_t^{\text{arm-local}}$ can be concatenated by different combinations of left or right or multiple body parts.

**3. World XYZ end-effector position condition (absolute)** Use absolute world-frame EE positions without any dataset or scale normalization:

$$\mathbf{c}_t^{\text{wXYZ}} = \bigoplus_{j \in J}p_t(j),$$

$$\Phi_{t:T}^{\text{wXYZ}} = \frac{1}{T}\sum_{u=t}^{t+T-1}\mathbf{c}_u^{\text{wXYZ}}.$$

**4. World XY root position condition (path control)** Use the root world-frame planar position. Remove an initial offset (for relative pathing) but do *not* normalize:

$$\mathbf{c}_t^{\text{wXY}} = \big(p_t^{\text{root}}\big)_{x,y} - \big(p_{t_0}^{\text{root}}\big)_{x,y},$$

$$\Phi_{t:T}^{\text{wXY}} = \frac{1}{T}\sum_{u=t}^{t+T-1}\mathbf{c}_u^{\text{wXY}}.$$

**5. World Z (root height) condition (absolute)**  Use the root world-frame height directly:

$$\mathbf{c}_t^{\mathrm{wZ}} \;=\; \big(p_t^{\mathrm{root}}\big)_z,$$

$$\Phi_{t:T}^{\mathrm{wZ}} \;=\; \frac{1}{T} \sum_{u=t}^{t+T-1} \mathbf{c}_u^{\mathrm{wZ}}.$$

**Notation addendum**

- $\bigoplus$: feature-wise concatenation along the channel/feature axis (we use $\bigoplus$ as "concat," not as a direct-sum operator).

- $\log(\cdot)$ on $SO(3)$: matrix logarithm mapping a rotation to a skew-symmetric element in $\mathfrak{so}(3)$.

- $\mathrm{vee}(\cdot)$: the $\vee$-operator $\mathfrak{so}(3) \to \mathbb{R}^3$ that converts a skew-symmetric matrix to its axial vector (inverse of the hat map).

- $\mathrm{wrap}(\cdot)$: angle wrapping to $(-\pi, \pi]$ (used for yaw increments).

- $\psi_t$: root yaw angle at time $t$; $(\mathbf{v}_t)_{x,y}$ and $(\cdot)_z$ denote planar and vertical components respectively.

- $J$: the chosen set of end-effectors/joints used for a condition (e.g., {wrist, elbow}); $a_j$ is the anchor joint for $j$ (e.g., shoulder for arm).

- $\ell_{a_j \to j}$: limb length from anchor $a_j$ to end-effector $j$ (measured in a rest pose or as an average over the dataset); $\varepsilon > 0$ is a small constant for numerical stability.

- $t_0$: the reference frame index used for offset removal in world-XY paths (typically the start of the current window, i.e., $t_0 = t$).

**Notes.**  Local (anchor-relative) features are made approximately scale-invariant by limb-length normalization, which is desirable for upper-limb control. World-frame conditions intentionally keep their physical magnitudes, preserving absolute task constraints such as global positions, heights, and true velocities. All conditions are aggregated to window-level features via $\Phi_{t:T}^{(\cdot)}$ to match the windowed formulation used in Sec. 3.3.

## A.5 DATASET PREPARATION AND PREPROCESSING

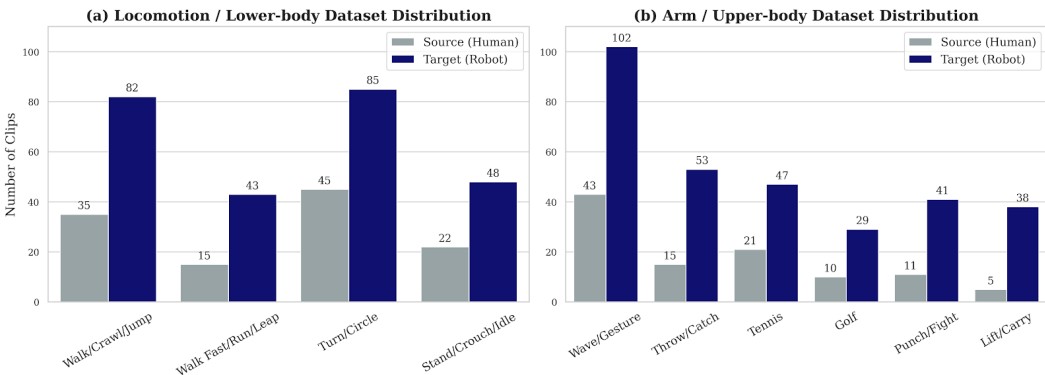

Figure 5: **Detailed distribution of motion clips used in experiments.** The dataset is categorized into (a) Locomotion and (b) Upper-body tasks. The target domain (Navy) consistently includes a broader set of motions across all categories to ensure diverse candidate coverage for the generative flow.

To ensure reproducibility and clarify our experimental setup, we detail the composition of our dataset and the preprocessing pipeline used to generate target domain motions.

**Dataset Distribution.** We source our motion data from the AMASS dataset (Mahmood et al., 2019), selecting subsets that cover distinct task types 4.1. As shown in Figure 5, we curate the dataset into two primary categories: Locomotion (lower-body dominant) and Upper-body manipulation. Crucially, we intentionally construct a larger and more diverse set of motions for the *Target (Robot)* domain compared to the *Source (Human)* domain (e.g., 310 target clips vs. 105 source clips for upper-body tasks). This asymmetry, visualized across detailed motion categories in Figure 5, is designed to ensure that the target codebook covers a broad manifold of potential motions—ranging from varying walking speeds to diverse reaching angles—providing our Flow Matching model with a rich set of candidates to satisfy various user conditions (e.g., local style vs. global task alignment).

**Preprocessing Pipeline.** Since AMASS provides human motion parameters (SMPL), we employ a retargeting-based preprocessing pipeline to adapt these motions to our target morphologies (Half-sized SMPL, Booster T1, and Spot) for training the target-side VQ-VAEs. This process consists of two stages:

- **Standardization:** We first extract standard joint positions and rotations from the raw AMASS archives. This step unifies the diverse optical marker data in AMASS into a consistent SMPL-based kinematic representation suitable for physics simulators.
- **Kinematic Adaptation via MINK:** To generate valid training data for non-humanoid or different-sized characters, we employ MINK (MuJoCo Inverse Kinematics) (Zakka, 2025), a python-based inverse kinematics solver. For each target robot (e.g., Spot), we define correspondence constraints mapping key human end-effectors (wrists, ankles) to the robot's equivalent links. We then run batch IK optimization to solve for the robot's joint configurations that best match the source human poses while respecting the robot's kinematic limits (e.g., joint ranges and limb lengths).

This preprocessing generates a large database of physically feasible motions for each target robot. These retargeted motions serve as the "ground truth" for training the target character's VQ-VAE tokenizer, effectively defining the latent motion space that MoReFlow learns to map into.

## A.6 LLM USAGE

We used a large language model solely to aid and polish writing (grammar, phrasing, and clarity); it did not contribute ideas, analyses, or results, and the authors take full responsibility for all content.

