# OpenReview forum: "MoReFlow: Motion Retargeting Learning through Unsupervised Flow Matching"
_ICLR.cc/2026/Conference — Submitted to ICLR 2026_

### Official Review · Reviewer_rBVb · 2025-10-27

**Soundness:** 3
**Presentation:** 4
**Contribution:** 3
**Rating:** 8
**Confidence:** 5

**Summary:**

The paper proposes retargeting motions between significantly different skeletal structures by conditioned flow matching in the motion tokens. Specifically, the motion tokenizer for each character is learned with VQ-VAE. Then, the flow matching conditioned by the motion-specific objective learns the motion token mapping from one character's motion to another. The conditions demonstrated are root velocity, local end effector position, world end effector position, world path control, and world root height control. The conditioned flow matching is trained with a similar strategy as classifier-free guidance by randomly dropping the condition with a probability. The demonstrations showcase challenging scenarios, such as retargeting humanoid motions to Spot (a quadruped robot equipped with a robotic arm on its head), which requires interpreting the body part mapping from the humanoid to the robotic arm.

**Strengths:**

* Simple and sensible design (VQVAE + conditioned flow matching) to achieve the goal
* Honest disclosure of limitations
* Clear and concise presentation with interesting demo scenarios, such as mapping humanoid motions to Spot and a chain-of-retargeting

**Weaknesses:**

* (minor) "Conditioned flow matching" rather than "unsupervised flow matching" may be a clearer way to describe the method without confusion
* As the authors infer in the limitations section, there is a scalability issue with this approach
  * There must be enough motions in each skeletal morphology to train sensible VQVAE tokenizers. This alone is challenging
  * Then, the conditioned flow matching must be trained with all possible conditions
  * The condition objectives must be hand-crafted. Five conditions predefined in the paper may not be enough for different motions, e.g., dancing

**Questions:**

* 8 hours of discrete flow retargeter training time - is this for training one conditioned flow from one morphology to another, or is this for all five conditions from one morphology to another?
* Related to the above training time and the weakness, any ideas on improving the scalability and the training speed?

---

> ### Author Response · Authors · 2025-11-21
> **First response, Regarding Reviewer's (rBVb) Comments**
>
> We sincerely appreciate the reviewer’s careful and understanding assessment of our contribution. Here we want to answer for your questions.
>
> **Question 1:** 8 hours of discrete flow retargeter training time - is this for training one conditioned flow from one morphology to another, or is this for all five conditions from one morphology to another?
>
> **A:** Based on our experiment, all five conditions from one morphology to another, takes 8 hours. Through your question, we realized that Section 4.2 may cause confusion for readers. We will therefore clarify training time more explicitly in Section 4.2.
>
> **Question 2:** Related to the above training time and the weakness, any ideas on improving the scalability and the training speed?
>
> **A:** Thank you for raising these critical points regarding scalability and training efficiency.
> - Improving Training Speed:
>
>     As you insightfuly noted, training speed and convergence are closely tied to the dataset distribution. Including a wide variety of motions indeed helps the Flow model find the optimal mapping path (flow trajectory) more easily. For example, a target dataset with a dense coverage of speeds allows the root-velocity conditioned flow to converge significantly faster by providing clear semantic correspondences.
>
> - Idea for Improving Scalability:
>
>     Regarding the scalability limitations you mentioned (such as data requirements per morphology), we acknowledge these constraints. However, we prioritized this design to ensure high-fidelity controllability, as "one-size-fits-all" approaches often compromise the precise physical alignment required for robotics in favor of scalability.
>
>     To specifically address the bottleneck of hand-crafted conditions for complex motions (e.g., dancing), we see a promising path in data-driven condition encoders. Instead of manually defining feature functions (as in Appendix A.4), future work could leverage pre-trained embeddings to automatically define $\Phi(x, c)$. This would allow the flow model to align motions based on high-level semantics without manual feature engineering, significantly improving the framework's extensibility.

---

> > ### Comment · Reviewer_rBVb · 2025-11-26
> >
> > I thank the authors for their respectful responses to all reviews and their comments. I have no further questions.
> >
> > While there are remaining practical issues and limitations with the scalability (as in this discussion) and the temporal alignment (as in the discussions with the reviewer 2E3m) to the proposed setup, motion retargeting with flow matching of motion tokens is worth sharing with the community for further exploration. Therefore, I remain supportive of this paper.

---

### Official Review · Reviewer_KJNg · 2025-11-01

**Soundness:** 3
**Presentation:** 2
**Contribution:** 3
**Rating:** 6
**Confidence:** 3

**Summary:**

This paper proposes a motion retargeting method based on flow matching. For each pair of character skeletons, this framework first trains a motion tokenizer for each of the characters, thus encoding their motion sequences into a series of character-specific tokens. These tokens form a character-specific token space, on which the flow matching model is trained to learn an invertible mapping between these two spaces. Once trained, motions from one character can then be retargeted to another character through this framework. Experiments reported better performance compared to baseline methods.

**Strengths:**

This work demonstrates an interesting framework for motion retargeting, which is shown to work between characters with different skeleton structures. From my perspective, the main strengths of this work include:
- A motion retargeting framework supports multiple conditions. I’m more in the character animation area, so it’s the first time for me to know that motion retargeting has a higher task-specific requirement in robotics. By supporting different conditions during formulation, this framework can adapt to different scenarios depending on the specific tasks.
- A motion retargeting framework trained with unpaired data. I think this is another interesting part of this paper, which alleviates the training data requirement for motion retargeting.

**Weaknesses:**

[Ablation study]
- For ablation study, I would expect to see some qualitative results as the current reported results are quite hard to evaluate. Only a limited number of metric values are reported, making the conclusions not so convincing to me.

[Paper writing]
- As I wrote in the Strengths section, I feel the unpaired training strategy is a critical and interesting contribution of this work. However, the details of this part are mainly put in supplemental, and the current description in Section 3.3 is a little bit hard to understand. I feel it would be better if the authors can elaborate more in this part, and potentially arrange some space to put the pseudo code in the main paper.

**Questions:**

I don’t have specific questions regarding this manuscript. Overall this is an interesting work, but the lack of qualitative results and potential improvement in the paper writing (see the Weaknesses section) restricts me from giving a higher score on it.

---

> ### Author Response · Authors · 2025-11-20
> **First response, Regarding Reviewer's (KJNg) Comments**
>
> We sincerely appreciate for your review. Below we address your concerns and answer questions. And we want to present changes will be made to the paper.
>
> **Weakness 1:** For ablation study, I would expect to see some qualitative results as the current reported results are quite hard to evaluate. Only a limited number of metric values are reported, making the conclusions not so convincing to me.
>
> **A:** To address your concern, we improved the ablation sections by including new summarized table in our paper.
>
> | Configuration        | FID ↓ | DIV ↑ | ALI ↑ | NAT (%) ↑ |
> |----------------------|-------|--------|--------|------------|
> | Continuous Latent    | 53.3  | **0.43** | **0.65** | 81.8       |
> | **Discrete (Ours)**  | **36.1** | 0.68 | 0.83 | **91.1**   |
>
> | Window Size        | ALI ↑ | NAT (%) ↑ |
> |--------------------|-------|------------|
> | **32 Frames (Default)** | **0.83** | 91.1       |
> | 64 Frames          | 0.81  | **92.5**   |
>
> | Coupling Batch Size | NAT (%) ↑ |
> |---------------------|------------|
> | 128 (Small)         | 89.0       |
> | **512 (Optimal)**   | **91.1**   |
> | 1024 (Large)        | 82.8       |
>
> To enhance readability, we decided to add a summary tables for Impact of Latent Space Design, Window Size, and Coupling Batch Size, which supporting illustration of ablation context.
>
> **Weakness 2:** As I wrote in the Strengths section, I feel the unpaired training strategy is a critical and interesting contribution of this work. However, the details of this part are mainly put in supplemental, and the current description in Section 3.3 is a little bit hard to understand. I feel it would be better if the authors can elaborate more in this part, and potentially arrange some space to put the pseudo code in the main paper.
>
> **A:** Thanks for your suggestion, and we sincerly agree with your opinion. We will improve and add technical details in Section 3.3 and Appendix to explain Multi-Sample Condition Coupling as well as a diagram to our method section.

---

### Official Review · Reviewer_2E3m · 2025-11-01

**Soundness:** 2
**Presentation:** 2
**Contribution:** 3
**Rating:** 2
**Confidence:** 5

**Summary:**

The flow matching is introduced in this manuscript to achieve motion retargeting in latent feature space. First, the motion sequences of each character are used to train a invididual  set of encoder, decoder and codebook of VQ-VAE for each character. After that, flow matching is involved to learn the mapping between the latent space distribution of different skeletons for each non-homeomorphic skeleton pairs. During training, the learned motion retargeting mapping function is introduced to achieve the retargeting between latent motion vector of different characters. The training of the mapping function is unsupervised without paired motion data under different skeletons. The proposed method is evaluated on the AMASS dataset to show its effectiveness.

**Strengths:**

1. The idea of using flow motion to achieve motion retargeting in the tokenized motion space is novel. This manuscript is a good pratices for applying VQ-VAE in the motion retargeting task;

2. The proposed method achieves good performance compared with existing method in both motion fidelity and controllability;

**Weaknesses:**

1. The setting of training an individual set of an encoder, a decoder and a codebook for each character seems not reasonable of at least not efficient. Considering that the motion retargeting mapping function is trained in the latent token space, it would be better to learn the codebook of each characters in a shared latent token space rather than in distinct spaces. Therefore, all the characters need to share the encoder and decoder. The difference between these two choices need further discussion;

2. The learning of motion retargeting in the latent motion space without paired data is not reliable especially when the set of motion category is large and the type of these categories is diverse;

3. The presentation of this manuscript is not good. There should be tables or figures to show the experimental results, not just by text descriptions;

4. As the author mentioned, the motion retargeting mapping model can only be applied on characters that are seen during training. For those characters that is unseen in advance, the retargeting will fail;

**Questions:**

1. The detail of the condition variable c is not clear. It is a one-hot vector. What is the set of the category the c support? How to indicate the condition during training for example how to specify the style condition and how ensure the c help the retargeting function learn the style transfer;

2. Please expain the reason why the encoder, decoder and codebook are learned for each of the characters? Why not share the encoder and decoder for all the characters. If the encoder and decoder is not shared, how to make sure that the latent motion space is shared?

3. Is there any trick for the selection of batch during training? I am wondering if the motion sequence of the source and target characters are quite different from each other, it is not easy for the learning of the retargeting fucntion to converge;

**Details Of Ethics Concerns:**

No concern.

---

> ### Author Response · Authors · 2025-11-20
> **First response(1), Regarding Reviewer's (2E3m) Comments**
>
> We are grateful for your valuable review. We have carefully revisited our design based on your feedback. In the subsequent sections, we address the concerns and questions you raised.
>
> **Weakness 1 & Question 2:** Why are the encoder, decoder, and codebook learned individually for each character rather than sharing them to share a common latent space? Given that a shared architecture seems more efficient and reasonable for retargeting, how does your separate design ensure that the latent motion spaces are shared/aligned without sharing the encoder and decoder?
>
> **A:** We agree that a shared latent space is a common choice for homogeneous domains. A shared encoder/decoder for different embodiments is a structure been adopted many previous work (e.g., SAME [1]). However, in our cross-embodiment setting (Human → Quadruped), enforcing a single shared space creates a severe bottleneck due to fundamental kinematic differences. To support our claim, we conduct a study on motion quality between different design decision:
>
> We unified input dimensions by applying zero-padding and masking to joint positions/velocities. While we employed these standard best practices to facilitate learning, this forces the network to process a sparse and topologically inconsistent input space.
>
> *This table solely evaluates Stage 1 (Representation). While the WtD-style shared codebook reconstructs locomotion well, its reliance on a "phase manifold" introduces a strong periodic bias, limiting its applicability to non-periodic tasks and leading to inferior retargeting performance as shown in our main results (Table 1).*
>
> | Method                                           | Reconstruction MSE (↓) | NAT (Human) | NAT (Spot) |
> |--------------------------------------------------|--------------------------|-------------|------------|
> | Individual Enc/Dec + Individual codebook (ours) | **0.00363515 (Avg)**        | **95.4%**       | **90.1%**      |
> | shared Enc/Dec + shared codebook                | 0.0566914                | 73.3%       | 34.7%      |
> | Individual Enc/Dec + shared codebook (WtD)      | 0.0029529 (Avg)          | 92.5%       | 90.4%      |
>
> As shown in the table, the shared model degrades massively. The Reconstruction MSE is ~15x higher than ours, and the Naturalness (NAT) for Spot drops to 34.7%. This confirms "Representation Collapse", a single network cannot simultaneously approximate the distinct manifolds of a biped and a quadruped, resulting in broken, floating, or sliding motions. Even the SOTA method Walk-the-Dog [2] employs separate encoders for this very reason. Furthermore, our modular designed decision offers superior flexibility: adding a new robot (e.g., Hexapod) requires training only one small VQ-VAE (<1 hour) without retraining the entire system, which is more adaptive to industrial application.
>
> We agree that this design choice and its alternatives are important to discuss explicitly. In the revised version, we will add a paragraph in the Method section.
>
> [1] Sunmin Lee, Taeho Kang, Jungnam Park, Jehee Lee, and Jungdam Won. Same: Skeleton-agnostic
> motion embedding for character animation. In SIGGRAPH Asia 2023 Conference Papers, pp.
> 1–11, 2023.
>
> [2] Peizhuo Li, Sebastian Starke, Yuting Ye, and Olga Sorkine-Hornung. Walkthedog: Crossmorphology motion alignment via phase manifolds. In SIGGRAPH, Technical Papers, 2024a.
>
> **Question 1:** The detail of the condition variable c is not clear. It is a one-hot vector. What is the set of the category the c support? How to indicate the condition during training for example how to specify the style condition and how ensure the c help the retargeting function learn the style transfer;
>
> **A:** $c$ is a one-hot vector representing an alignment objective, not a motion category. The supported categories, detailed in Appendix A.4, include objectives such as "Local Pose Alignment" (for style preservation), "World End-Effector Alignment" (for task precision), and "Root Velocity Alignment."
>
> During training, $c$ acts as a switch that guarantees the desired behavior. It is (1) fed directly into the flow network to condition the generation, and (2) used to select the corresponding feature function $\Phi(x, c)$ for calculating the feature loss $\mathcal{L}_{feat}$. For instance, when $c$ indicates "style," the loss explicitly minimizes the distance in local pose features. This mechanism forces the retargeting function to learn the specific transfer logic (e.g., preserving style vs. global position) associated with each condition.

---

> > ### Comment · Reviewer_2E3m · 2025-11-22
> >
> > What about the method "shared Enc/Dec with individual codebook"? How is the performance?

---

> ### Author Response · Authors · 2025-11-20
> **First response(2), Regarding Reviewer's (2E3m) Comments**
>
> **Weakness 2 & Question 3:** How is the learning of motion retargeting in the latent space reliable without paired data, especially when the set of motion categories is large and diverse? Is there any specific 'trick' for batch selection during training to ensure convergence when the source and target motion sequences differ significantly?
>
> **A:** Yes, there are specific mechanisms designed to ensure convergence and reliability: 1) Multi-Sample Condition Coupling and 2) Feature Loss $\mathcal{L}_{feat}$. To address the concern about reliability on diverse categories, we do not attempt to match the "entire source distribution" to the "entire target distribution" globally, which would indeed be unstable. Instead, we align conditional distributions $p(z|c)$. By fixing a condition $c$ (e.g., root velocity), we ensure that the model only couples and learns from source/target chunks that are semantically comparable within that specific subset.
>
> To address your specific question about "motion sequence differences," we emphasize that we do not match long, unstructured sequences. As detailed in our implementation, we process motions in fixed short windows (e.g., 32 frames), which the VQ-VAE compresses into latent sequences of just 8 tokens. This means the "sample" in our coupling is a short local window, not a full trajectory. This significantly aids convergence because the model only needs to align local kinematic patterns rather than complex long-term dynamics.
>
> Within these short windows, our Multi-Sample Condition Coupling strategy acts as the effective "trick" for batch selection 2. We sample source and target batches independently, compute a cost matrix based on the condition-specific feature $\Phi(x, c)$, and dynamically construct pseudo-pairs (e.g., pairing a "fast source window" with a "fast target window"). The Flow Matching loss $\mathcal{L}\_{FM}$ learns the mapping between these coupled tokens, while the feature loss $\mathcal{L}\_{feat}$ ensures the decoded motion physically matches the source in the feature space.
>
> In summary, the "trick" is the combination of short-window processing and condition-guided coupling. Even when raw motion sequences differ significantly in joint space, this mechanism ensures optimization is always grounded in comparable condition features. We will revise Section 3.3 to strictly detail this rationale, demonstrating why this design makes unsupervised retargeting reliable even when motion categories are large and diverse.
>
> **Weakness 4:** As the author mentioned, the motion retargeting mapping model can only be applied on characters that are seen during training. For those characters that is unseen in advance, the retargeting will fail;
>
> **A:** We agree with this observation and explicitly acknowledge it as a limitation in Section 5. Our current design deliberately focuses on high-fidelity, controllable retargeting for specific, extremely heterogeneous morphology pairs (e.g., Booster T1 ↔ human ↔ Spot), and thus learns one-to-one mappings only for characters seen during training. In this regime, we prioritize precise semantic alignment and stability over any-to-any generalization across arbitrary new skeletons. Extending MoReFlow to handle unseen characters is an important and nontrivial direction that we view as future work, and we believe the analyses in our paper provide a useful foundation for such extensions.

---

> > ### Comment · Reviewer_2E3m · 2025-11-22
> >
> > If the unsupervised matching is applied on short-windowed motion, temporal alignment is further needed for training covergence, just as the authors mention in the rebuttal, namely pairing a "fast source window" with a "fast target window". Simple temporal alignment based on for example motion velocity may be enough when the temporal complexity of the motion sequences is low. However in real application, simple temporal alignment may result in mismatch between the source short-window motion and the target short-window motion, and the training of the motion retargeting function will fail. Temporal alignment bewteen two motion sequences from differenct actors is still a challenging problem.

---

> > > ### Author Response · Authors · 2025-11-24
> > >
> > > **Q:** However in real application, simple temporal alignment may result in mismatch between the source short-window motion and the target short-window motion, and the training of the motion retargeting function will fail. Temporal alignment bewteen two motion sequences from differenct actors is still a challenging problem.
> > >
> > > **A:**
> > > 1. Acknowledging the Challenge & Our Contribution:
> > >
> > >     We fully agree that temporal alignment between different actors is a compound and challenging problem—one that prior unsupervised methods have struggled to address effectively without paired data. We believe our work advances the solution to this problem precisely by incorporating explicit conditional terms ($c$) to achieve robust alignment. Unlike previous methods that might rely on implicit or simple heuristics, our framework allows the alignment objective to be adjustable and constrained by specific physical definitions.
> > >
> > >
> > >
> > > 2. Beyond "Simple" Temporal Alignment:
> > >
> > >     The reviewer is correct that simple alignment (e.g., scalar velocity) is insufficient for complex motions. However, our method deliberately avoids relying on such simple scalars or global frame-by-frame matching.
> > >
> > >     - High-Dimensional Features: Instead, for a fixed condition $c$, we compute rich condition features $\Phi(x, c)$ (e.g., task-space end-effector paths, height/trajectory cues, root-velocity profiles), as detailed in Appendix A.4.
> > >     - Short-Window Distributional Coupling: We operate on short local windows (32 frames) where these average features retain strong phase-discriminative information. Within a mini-batch, our coupling strategy uses these high-dimensional features to find the optimal semantic match. This mechanism prevents the mismatch you concerned; for example, a "jump takeoff" and "jump landing" might share similar velocities, but their EE position profiles $\Phi$ are distinct, preventing them from being incorrectly paired.
> > >
> > > 3. Empirical Robustness:
> > >
> > >     While we agree that any unsupervised approach has limits if motion coverage is non-overlapping, in our experimental setting (diverse locomotion and upper-body motions), this condition-based local coupling did not lead to training failure. On the contrary, we observed stable convergence and consistent improvements over baselines.
> > >
> > > We will clarify in Section 3.3 that our goal is a practical distributional matching scheme that works well under realistic cross-embodiment datasets, rather than solving the generalized temporal alignment problem for arbitrary sequences.

---

> ### Author Response · Authors · 2025-11-20
> **First response(3), Regarding Reviewer's (2E3m) Comments**
>
> **Weakness 3:** The presentation of this manuscript is not good. There should be tables or figures to show the experimental results, not just by text descriptions;
>
> **A:** We assume that the “presentation” you pointed out is about ablation section. This is because there are Result figure 1,2 and baseline table 1 for illustrating corresponding sections.
> To address your concern, we improved the ablation and baseline sections by including new summarized table and detailed result analysis of baslines in our paper.
> | Method            | FID ↓ Loco. | FID ↓ Upper. | DIV ↑ Loco. | DIV ↑ Upper. | ALI ↑ Loco. | ALI ↑ Upper. | NAT (%) ↑ Loco. | NAT (%) ↑ Upper. |
> |-------------------|-------------|---------------|--------------|---------------|--------------|----------------|-------------------|--------------------|
> | WtD-human         | 42.3        | 77.1          | 0.44         | 0.39          | 0.82         | 0.57           | 86.0              | 81.8               |
> | ACE-human         | 50.4        | 61.1          | 0.54         | 0.52          | 0.77         | 0.73           | 82.4              | 84.2               |
> | **MoReFlow-human** | **32.4**    | **35.8**      | **0.69**     | **0.64**      | **0.87**     | **0.80**       | **89.9**          | **93.5**           |
> | WtD-non-human     | 42.5        | 72.9          | 0.46         | 0.41          | **0.85**     | 0.54           | 88.4              | 82.6               |
> | ACE-non-human     | 53.0        | 60.6          | 0.58         | 0.50          | 0.80         | 0.77           | 79.3              | 84.0               |
> | **MoReFlow-non-human** | **33.8** | **42.0**     | **0.68**     | **0.62**      | 0.83         | **0.79**       | **89.2**          | **92.3**           |
>
> This newly divided result table based on human vs non-human criteria shows the same overall trend as before, but WtD-non-human exhibits a higher ALI-locomotion metric value than ours. This suggests that the retargeted motion for Spot demonstrated comparable or superior feature-level consistency in WtD. However, as noted in the manuscript, while WtD's phase-manifold design is effective for periodic gaits, MoReFlow achieves significantly higher motion realism (NAT) and fidelity (FID). We will include this new-observation in baseline section.
>
> | Configuration        | FID ↓ | DIV ↑ | ALI ↑ | NAT (%) ↑ |
> |----------------------|-------|--------|--------|------------|
> | Continuous Latent    | 53.3  | **0.43** | **0.65** | 81.8       |
> | **Discrete (Ours)**  | **36.1** | 0.68 | 0.83 | **91.1**   |
>
> | Window Size        | ALI ↑ | NAT (%) ↑ |
> |--------------------|-------|------------|
> | **32 Frames (Default)** | **0.83** | 91.1       |
> | 64 Frames          | 0.81  | **92.5**   |
>
> | Coupling Batch Size | NAT (%) ↑ |
> |---------------------|------------|
> | 128 (Small)         | 89.0       |
> | **512 (Optimal)**   | **91.1**   |
> | 1024 (Large)        | 82.8       |
>
> For ablation section, to enhance readability, we added summary tables for Impact of Latent Space Design, Window Size, and Coupling Batch Size, which supporting illustration of ablation context.

---

> ### Author Response · Authors · 2025-11-24
>
> **Q:** What about the method "shared Enc/Dec with individual codebook"? How is the performance?
>
> **A:** Thank you for sharing this interesting idea. We agree that the “Shared Enc/Dec with individual codebook” configuration is indeed intriguing. However, we would appreciate further clarification since this setup is not a standard design. Typically, architectures either share the full representational stack to enforce a common feature space or separate them to handle distinct topological constraints. Sharing the network weights (Enc/Dec) while strictly enforcing disjoint discrete spaces (individual codebooks) creates a design conflict that is rarely adopted.
>
> Could you kindly point us to specific reference papers or describe the exact implementation you have in mind? We want to ensure that any implementation we consider aligns perfectly with your vision. We anticipate this setup would still suffer from the same "kinematic interference" bottleneck caused by the shared encoder/decoder weights, similar to presented extra experiment we evaluated, however, we will be happy to explore.

---

> ### Author Response · Authors · 2025-12-04
> **Final Summary of discussion with Reviewer 2E3m by Authors**
>
> As the discussion period concludes, we would like to provide a final summary of how insightful discussion regarding architecture and temporal alignment have been addressed and incorporated into the revised manuscript.
>
> 1. Architectural Consensus: Why character-specific design is Necessary
>
>     Throughout our discussion, we explored various architectural configurations, including the reviewer’s initial suggestion of a fully shared space and the follow-up inquiry about "Shared Enc/Dec + Individual Codebooks."
>
>     - **Evidence:** As demonstrated in our additional experiment (Response 1), enforcing shared encoder/decoder weights across heterogeneous morphologies (Human $\rightarrow$ Quadruped) causes a **15-fold increase in MSE** and **Representation Collapse** due to kinematic bottlenecks.
>     - **Conclusion:** While the 'Shared Enc/Dec + Individual Codebook' configuration presents an interesting theoretical variant, it remains an unconventional architectural choice in cross-morphology retargeting.
>     Standard approaches typically either
>         - share the full representational stack to enforce a common feature space for homeomorphic skeletons (e.g., SAME [1]. via skeletal pooling to a primal skeleton; NKN [2]. via FK-layer with same topology assumption)
>         - decouple the encoder-decoder to respect distinct topological constraints.
>             - HuMoT [3]. To support variation in both morphology and topology, the encoding and decoding processes are conditioned on skeleton-specific templates. The model explicitly handles skeletal topology and learns a topology-agnostic motion representation.
>             - M-R^2ET [4]. This modular system separates a cross-structure alignment module from a residual modification module to preserve motion semantics across skeletons with different structures.
>             - T2M-X [5]. It trains three separate VQ-VAEs for the body, hands, and face, demonstrating that using distinct codebooks for kinematically different body parts significantly improves motion quality.
>
>         The suggested hybrid design from the reviewer introduces a structural conflict—enforcing shared weight optimization for disjoint discrete manifolds—which retains the fundamental bottleneck of kinematic incompatibility, as the shared encoder cannot simultaneously optimize for divergent joint hierarchies and kinematic structures.
>
>     - **Revision:** Based on this, we have finalized the **"Design Rationale" in Section 3.1**, explicitly stating that a decoupled architecture is structurally necessary to avoid the interference observed in our experiments.
>
>
>
> 2. Resolution on Temporal Alignment:
>
>     We deeply appreciate the reviewer’s scrutiny regarding the risks of "mismatched pairing" in short windows.
>
>     - **Clarification:** As clarified in our second response, MoReFlow avoids this pitfall by not relying on simple scalar metrics. Instead, our coupling utilizes **high-dimensional condition features** (including end-effector paths and phase-sensitive trajectory profiles).
>     - **Revision:** To prevent future misunderstanding, we have updated **Appendix A.4** and **Section 3.3** to explicitly detail these features, confirming that our distributional coupling is robust against the temporal ambiguities.
>
>
>
> 3. Comprehensive Improvements:
>
>     In addition to the above, we have addressed the reviewer’s other concerns by:
>
>     - Defining the condition variable $c$ in **Section 3.2**.
>     - Replacing text descriptions with **Figure 4 (Bar plots)** and **Table 1 (Detailed breakdown)**.
>
> We believe these revisions transform the reviewer’s critiques into significant improvements in the paper's rigor. We hope this final summary confirms that the reviewer’s concerns have been fully resolved.
>
> [1] Sunmin Lee, Taeho Kang, Jungnam Park, Jehee Lee, and Jungdam Won. Same: Skeleton-agnostic motion embedding for character animation. In SIGGRAPH Asia 2023 Conference Papers, pp. 1–11, 2023.
>
> [2] R. Villegas, J. Yang, D. Ceylan and H. Lee, "Neural Kinematic Networks for Unsupervised Motion Retargetting," 2018 IEEE/CVF Conference on Computer Vision and Pattern Recognition, Salt Lake City, UT, USA, 2018.
>
> [3] Mourot, L., Hoyet, L., Clerc, F. L., & Hellier, P. (2023). HuMoT: Human Motion Representation using Topology-Agnostic Transformers for Character Animation Retargeting. arXiv preprint arXiv:2305.18897.
>
> [4] J. Zhang, Z. Tu, J. Weng, J. Yuan and B. Du, "A Modular Neural Motion Retargeting System Decoupling Skeleton and Shape Perception," in IEEE Transactions on Pattern Analysis and Machine Intelligence, vol. 46, no. 10, pp. 6889-6904, Oct. 2024.
>
> [5] Liu, M., Liu, Y., Krishnan, G., Bayer, K. S., & Zhou, B. (2024). T2M-X: Learning Expressive Text-to-Motion Generation from Partially Annotated Data. arXiv preprint arXiv:2409.13251.

---

### Official Review · Reviewer_Khj3 · 2025-11-01

**Soundness:** 3
**Presentation:** 2
**Contribution:** 2
**Rating:** 4
**Confidence:** 4

**Summary:**

This paper proposes **MoReFlow**, a framework for **cross-embodiment motion retargeting**. The motivation stems from the limitations of existing approaches, which typically assume identical human morphologies or a fixed retargeting objective. MoReFlow introduces a **VQ-VAE-based tokenizer** to encode motions into a discrete latent space, combined with a **conditional flow matching network** for motion retargeting within this space. Furthermore, a **multi-sample coupling strategy** is designed to associate each source motion with multiple target motions, enabling unsupervised learning. Experiments on both upper-body and lower-body motions across four embodiments demonstrate the effectiveness of the proposed approach.

**Strengths:**

**Strengths:**

[Sound Motivation and Methodology]

- The paper clearly identifies the limitations of existing motion retargeting methods that are restricted to fixed morphologies or deterministic mapping objectives.
- The proposed conditional flow matching mechanism is a technically sound and promising approach to address these challenges.

[Advantages in Controllability]

- MoReFlow allows controllable retargeting styles (e.g., local vs. global), which significantly enhances its applicability and flexibility for different use cases.

[Ablation Study to Verify Effectiveness]

- The paper includes ablation analyses to verify the contributions of key design components, supporting the soundness of the proposed architecture.

**Weaknesses:**

**Weaknesses:**

[Incomplete Methodological Explanation]

- The conditioning feature vector is insufficiently described: it is only mentioned to differ between animation and robotics, without providing a general, conceptual definition. A clearer discussion in the main text (rather than relegated to the appendix) would help readers understand its role and composition.
- The rationale behind the multi-sample coupling strategy—a critical component for achieving unpaired learning—is not well articulated. The paper should better explain why and how this mechanism facilitates generalization since it is essential component for applying flow matching to motion retargeting.

[Assumptions on Cross-Embodiment Data]

- The method seems to require roughly comparable motion coverage across embodiments, even if motions are not paired. This assumption could limit scalability to non-humanoid or newly introduced embodiments where motion data is scarce.

[Insufficient Experiments]

- The experimental comparisons are relatively weak: only two baseline methods are considered, despite the existence of many relevant cross-embodiment retargeting approaches discussed in related work.
- Results are aggregated across embodiments, preventing detailed analysis (e.g., humanoid vs. non-humanoid performance).
- The origin and preprocessing of the evaluation datasets are not clearly described, which reduces reproducibility.

[Improvements to Writing]

- The paper could better highlight the novel contributions of MoReFlow in the related work section to emphasize its differentiation especially why it is necessary to apply flow matching compared to existing motion retargeting work.
- The ablation results are described only in text, making it difficult to assess their impact. Presenting them in a summary table would improve readability and support the claims more convincingly.

**Questions:**

Please refer to the problems discussed in weaknesses.

---

> ### Author Response · Authors · 2025-11-20
> **First response(1), Regarding Reviewer's (Khj3) Comments**
>
> We appreciate for your thoughtful review. We carefully revisited our design in light of your comments. Below we address your concerns and answer questions.
>
> **Weakness 1-1:** The conditioning feature vector is insufficiently described: it is only mentioned to differ between animation and robotics, without providing a general, conceptual definition. A clearer discussion in the main text (rather than relegated to the appendix) would help readers understand its role and composition.
>
> **A:** We want to clarify that the conditioning feature vector and the variable $c$ represent the alignment objective that guides how motions are compared and generated, rather than a dataset-specific label. In practice, cselects which metric the model should satisfy, such as matching local pose features or matching world-space task features, and the flow is trained to produce motions consistent with that chosen objective. To address your concern, we will revise Section 3.2 to provide a clear conceptual definition of $c$ and the conditioning features, explain how they operate during training and inference, and update Figure 3 so that their roles are immediately understandable from the main text.
>
> **Weakness 1-2:** The rationale behind the multi-sample coupling strategy—a critical component for achieving unpaired learning—is not well articulated. The paper should better explain why and how this mechanism facilitates generalization since it is essential component for applying flow matching to motion retargeting.
>
> **A:** The multi-sample coupling strategy facilitates generalization by acting as a stochastic regularization mechanism. Unlike fixed pairing which risks overfitting to specific target instances, our dynamic batch-wise coupling associates the same source motion with different semantically similar target motions across training steps (e.g., a source "Fast Run" pairs with various "Fast Gallop" targets across epochs). This "one-to-many" exposure acts as data augmentation, forcing the Flow model to learn a generalized semantic direction rather than memorizing specific trajectories.
>
> Furthermore, this mechanism injects a critical "weak supervision signal" into the unpaired setting. By solving for optimal assignments based on condition-feature distance $||\Phi(x^{src}, c) - \Phi(x^{tgt}, c)||$ , we ensure the model trains on "pseudo-pairs" that respect physical properties. This theoretical basis explains how the model avoids mode collapse and learns robust correspondences that generalize to unseen data, rather than learning arbitrary mappings. We will revise Section 3.3 and the Appendix to explicitly articulate this rationale with a new diagram to visually illustrate the process.
>
> **Weakness 2:** The method seems to require roughly comparable motion coverage across embodiments, even if motions are not paired. This assumption could limit scalability to non-humanoid or newly introduced embodiments where motion data is scarce.
>
> **A:** Thank you for your insightful thought. We want to clarify that our framework relies on two distinct assumptions. First, we agree with your observation that a newly introduced embodiment requires enough data to train a valid VQ-VAE tokenizer; this is a prerequisite to building its motion "vocabulary." However, for the retargeting stage itself, we do not assume roughly comparable motion coverage. In fact, handling mismatched coverage is a core design feature of our framework.
>
> Our "multi-sample condition coupling" strategy handles distributional discrepancies by dynamically finding the best available semantic match rather than requiring exact 1:1 pairings. For instance, our source dataset contains humans running at 2.0 m/s, while the target half-sized robot dataset lacks this speed due to physical limits. Instead of failing, our framework correctly maps the 2.0 m/s source motion to the fastest available running motion in the target distribution, ensuring the closest semantic equivalent is chosen.
>
> Therefore, regarding scalability, our approach remains robust to data scarcity as long as there is reasonable semantic overlap (e.g., any form of locomotion) between the distributions. By prioritizing best-effort correspondence over strict distributional matching, MoReFlow can generalize to new embodiments with scarce or incomplete data coverage more effectively than methods requiring strictly overlapping domains. We are happy to elaborate more throughly.

---

> ### Author Response · Authors · 2025-11-20
> **First response(2), Regarding Reviewer's (Khj3) Comments**
>
> **Weakness 3-1:** The experimental comparisons are relatively weak: only two baseline methods are considered, despite the existence of many relevant cross-embodiment retargeting approaches discussed in related work.
>
> **A:** We appreciate this concern and agree that, in general, stronger baselines make a paper more convincing. In our specific setting, however, the pool of directly comparable methods is much smaller than the full list of works discussed in Section 2. Our problem is unsupervised cross-embodiment motion retargeting in a shared offline dataset (AMASS), across strongly different morphologies, with a learned motion prior and no paired correspondences or task-specific rewards. Many prior approaches in the related work are (a) same-morphology human motion models [1], (b) optimization-based retargeting that assumes paired poses or hand-crafted constraints [2] or (c) RL / control pipelines that optimize task rewards in a simulator rather than learning a reusable retargeting map [3]. Under this stricter definition, WtD and ACE are, to our knowledge, the only recent SOTA methods that both (i) learn a motion prior and (ii) perform cross-character retargeting in an unsupervised way, which is why we selected them as our main baselines (Sec. 4.4).
> We will provide a more thorough justification in Section 4.4, clarifying why these two methods are the most directly comparable baselines in our experimental setting.
>
> [1] Ruben Villegas, Jimei Yang, Seunghoon Hong, Xunyu Lin, and Honglak Lee. Learning to generate
> long-term future via hierarchical prediction. In Proceedings of the 35th International Conference
> on Machine Learning, 2018.
>
> [2] T. Moulard, E. Yoshida and S. Nakaoka, "Optimization-based motion retargeting integrating spatial and dynamic constraints for humanoid," IEEE ISR 2013.
>
> [3] Tianyu Li, Hyunyoung Jung, Matthew Gombolay, Yong Kwon Cho, and Sehoon Ha. Crossloco: Human motion driven control of legged robots via guided unsupervised reinforcement learning.
>
> **Weakness 3-3:** The origin and preprocessing of the evaluation datasets are not clearly described, which reduces reproducibility.
>
> **A:** Based on your point, we also believe describing the information and preprocessing of dataset we have done is helpful for increasing reproducibility. We will additional Appendix section for describing what we have used from AMASS dataset and how we preprocessed the dataset in Appendix A.5.
>
> **Weakness 4-1:** The paper could better highlight the novel contributions of MoReFlow in the related work section to emphasize its differentiation especially why it is necessary to apply flow matching compared to existing motion retargeting work.
>
> **A:** In the revision, we will explicitly highlight why MoReFlow adopts conditional flow matching rather than more common GAN/VAE-based mappings: (i) unlike GANs, flow matching provides stable, likelihood-based training without adversarial failure modes; (ii) unlike standard VAEs, it learns a highly expressive, non-collapsed mapping that does not blur diverse motions into an “average” style; and (iii) the learned vector field defines a reversible, continuous path between source and target tokens, which makes it particularly well-suited for conditional control (e.g., switching between local/global conditions, speed, or embodiment) by conditioning the flow on $c$.
> Thank you for this suggestion, we will add a short paragraph at the end of Section 2.

---

> ### Author Response · Authors · 2025-11-20
> **First response(3), Regarding Reviewer's (Khj3) Comments**
>
> **Weakness 3-2:** Results are aggregated across embodiments, preventing detailed analysis (e.g., humanoid vs. non-humanoid performance).
>
> **A:** We sincerly consider your recommendation and we decided to follow your point that dividing the result into character-wise. Our work is working for all different sized humanoids and quadrupeds, so we also think it will be helpful to analysis results by dividing it in baseline table. We will include this revised table in revision paper.
> | Method            | FID ↓ Loco. | FID ↓ Upper. | DIV ↑ Loco. | DIV ↑ Upper. | ALI ↑ Loco. | ALI ↑ Upper. | NAT (%) ↑ Loco. | NAT (%) ↑ Upper. |
> |-------------------|-------------|---------------|--------------|---------------|--------------|----------------|-------------------|--------------------|
> | WtD-human         | 42.3        | 77.1          | 0.44         | 0.39          | 0.82         | 0.57           | 86.0              | 81.8               |
> | ACE-human         | 50.4        | 61.1          | 0.54         | 0.52          | 0.77         | 0.73           | 82.4              | 84.2               |
> | **MoReFlow-human** | **32.4**    | **35.8**      | **0.69**     | **0.64**      | **0.87**     | **0.80**       | **89.9**          | **93.5**           |
> | WtD-non-human     | 42.5        | 72.9          | 0.46         | 0.41          | **0.85**     | 0.54           | 88.4              | 82.6               |
> | ACE-non-human     | 53.0        | 60.6          | 0.58         | 0.50          | 0.80         | 0.77           | 79.3              | 84.0               |
> | **MoReFlow-non-human** | **33.8** | **42.0**     | **0.68**     | **0.62**      | 0.83         | **0.79**       | **89.2**          | **92.3**           |
>
> This newly divided result table based on human vs non-human criteria shows the same overall trend as before, but WtD-non-human exhibits a higher ALI-locomotion metric value than ours. This suggests that the retargeted motion for Spot demonstrated comparable or superior feature-level consistency in WtD. However, as noted in the manuscript, while WtD's phase-manifold design is effective for periodic gaits, MoReFlow achieves significantly higher motion realism (NAT) and fidelity (FID). We will include this new-observation in baseline section.
>
> **Weakness 4-2:** The ablation results are described only in text, making it difficult to assess their impact. Presenting them in a summary table would improve readability and support the claims more convincingly.
>
> **A:** To address your concern, we improved the ablation sections by including new summarized table in our paper.
>
> | Configuration        | FID ↓ | DIV ↑ | ALI ↑ | NAT (%) ↑ |
> |----------------------|-------|--------|--------|------------|
> | Continuous Latent    | 53.3  | **0.43** | **0.65** | 81.8       |
> | **Discrete (Ours)**  | **36.1** | 0.68 | 0.83 | **91.1**   |
>
> | Window Size        | ALI ↑ | NAT (%) ↑ |
> |--------------------|-------|------------|
> | **32 Frames (Default)** | **0.83** | 91.1       |
> | 64 Frames          | 0.81  | **92.5**   |
>
> | Coupling Batch Size | NAT (%) ↑ |
> |---------------------|------------|
> | 128 (Small)         | 89.0       |
> | **512 (Optimal)**   | **91.1**   |
> | 1024 (Large)        | 82.8       |
>
> To enhance readability, we added summary tables for Impact of Latent Space Design, Window Size, and Coupling Batch Size, which supporting illustration of ablation context.

---

### Author Response · Authors · 2025-11-20
**General Comments**

We sincerely appreciate all reviewers for the constructive feedback. Here we address the common concerns raised by the reviewers.

**1. Clarifying the “Multi-Sample Condition Coupling” strategy**
- **Q:** Reviewer **Khj3** asked for the rationale behind the unpaired learning strategy from. And, reviewer **2E3m** asked how the in-batch coupling mechanism works to ensure reliability without paired data.
- **A:** This strategy is the core mechanism that enables unsupervised learning, while avoiding the need for expensive manual motion pairing between heterogeneous robots. During the training, instead of random pairing, we sample batches independently and compute a cost matrix between all source-target pairs using the condition-specific feature $\Phi(x, c)$. We then solve for the optimal assignment to form "pseudo-pairs". This provides the necessary "weak supervision" to prevent mode collapse and ensures semantic consistency during training.

    As requested by Reviewer **KJNg**, We will improve our revision by adding technical details as well as a diagram to our method section and Appendix.

**2. Clarifying the Condition Variable $c$ and Feature Vector**
- **Q:** Reviewer **Khj3** pointed out the lack of a conceptual definition for the condition variable $c$ . And, Reviewer **2E3m** asked how it physically guides the retargeting process.
- **A:** Conceptually, $c$ represents the "alignment objective" (e.g., "Match Local Pose" vs. "Match World Task"), not just a motion label. It acts as a switch:
    - During Training: $c$ selects which feature function $\Phi(x, c)$ is used to calculate the loss. For example, if $c$="Style", the model minimizes the distance in *local joint positions*; if $c$="Task", it minimizes the distance in *world EE positions*.
    - During Inference: The flow model uses this learned relationship to generate motion that satisfies the specific objective $c$.

    We will improve Section 3.2 to explicitly define $c$ as the objective selector and $\Phi(x, c)$ as the metric, and also will be elaborated in the method diagram. We will also update the overview diagram (Figure 3) accordingly.


**3. Improving ablation presentation and result granularity**
- **Q:** Reviewers (**Khj3, 2E3m, KJNg**) pointed out that the experimental results, particularly the ablation studies, were described only in text, which made it difficult to assess the impact of design choices. Additionally, Reviewer **Khj3** requested a breakdown of performance by embodiment type (Humanoid vs. Non-Humanoid) rather than aggregated results.
- **A:** We agree that text-only descriptions reduced readability and aggregated baseline results masked the performance differences across different embodiments on retargeting tasks.

    We added a new summary table for the ablation studies to improve readability. Furthermore, we have reorganized Table 1 to explicitly separate results into "human(Original SMPL robot to SMPL-half robot)" and "non-human(Original SMPL robot to Spot)" categories to support detailed analysis.
[Baseline Table]
| Method            | FID ↓ Loco. | FID ↓ Upper. | DIV ↑ Loco. | DIV ↑ Upper. | ALI ↑ Loco. | ALI ↑ Upper. | NAT (%) ↑ Loco. | NAT (%) ↑ Upper. |
|-------------------|-------------|---------------|--------------|---------------|--------------|----------------|-------------------|--------------------|
| WtD-human         | 42.3        | 77.1          | 0.44         | 0.39          | 0.82         | 0.57           | 86.0              | 81.8               |
| ACE-human         | 50.4        | 61.1          | 0.54         | 0.52          | 0.77         | 0.73           | 82.4              | 84.2               |
| **MoReFlow-human** | **32.4**    | **35.8**      | **0.69**     | **0.64**      | **0.87**     | **0.80**       | **89.9**          | **93.5**           |
| WtD-non-human     | 42.5        | 72.9          | 0.46         | 0.41          | **0.85**     | 0.54           | 88.4              | 82.6               |
| ACE-non-human     | 53.0        | 60.6          | 0.58         | 0.50          | 0.80         | 0.77           | 79.3              | 84.0               |
| **MoReFlow-non-human** | **33.8** | **42.0**     | **0.68**     | **0.62**      | 0.83         | **0.79**       | **89.2**          | **92.3**           |

[Ablation Tables]
| Configuration        | FID ↓ | DIV ↑ | ALI ↑ | NAT (%) ↑ |
|----------------------|-------|--------|--------|------------|
| Continuous Latent    | 53.3  | **0.43** | **0.65** | 81.8       |
| **Discrete (Ours)**  | **36.1** | 0.68 | 0.83 | **91.1**   |

| Window Size        | ALI ↑ | NAT (%) ↑ |
|--------------------|-------|------------|
| **32 Frames (Default)** | **0.83** | 91.1       |
| 64 Frames          | 0.81  | **92.5**   |

| Coupling Batch Size | NAT (%) ↑ |
|---------------------|------------|
| 128 (Small)         | 89.0       |
| **512 (Optimal)**   | **91.1**   |
| 1024 (Large)        | 82.8       |

---

> ### Author Response · Authors · 2025-11-20
>
> **4. Validating character-specific VQ-VAEs Instead of one shared tokenizer (thanks to Reviewer 2E3m)**
> - **Q:** Reviewer **2E3m** suggested that a single shared encoder/decoder/codebook might simplify alignment and improve efficiency, questioning the necessity of per-character tokenizers.
> - **A:** Following the reviewers’ suggestion, we conducted an additional ablation experiment comparing: (i) our character-specific VQ-VAEs, (ii) a single shared-tokenizer, and (iii) a WtD-style shared-codebook baseline.
>
>     Under identical configurations, the shared-tokenizer significantly degraded reconstruction quality and NAT scores for both Human and Spot.
>
>     *This table solely evaluates Stage 1 (Representation). While the WtD-style shared codebook reconstructs locomotion well, its reliance on a "phase manifold" introduces a strong periodic bias, limiting its applicability to non-periodic tasks and leading to inferior retargeting performance as shown in our main results (Table 1).*
>
>     | Method                                           | Reconstruction MSE (↓) | NAT (Human) | NAT (Spot) |
>     |--------------------------------------------------|--------------------------|-------------|------------|
>     | Individual Enc/Dec + Individual codebook (ours) | **0.00363515 (Avg)**        | **95.4%**       | **90.1%**      |
>     | shared Enc/Dec + shared codebook                | 0.0566914                | 73.3%       | 34.7%      |
>     | Individual Enc/Dec + shared codebook (WtD)      | 0.0029529 (Avg)          | 92.5%       | 90.4%      |
>
>     This confirms that strongly heterogeneous morphologies are difficult to be cleanly represented in one latent space, whereas per-character tokenizers better preserve each character’s motion manifold. We will add a concise subsection in Methods summarizing this design choice.

---

### Author Response · Authors · 2025-12-04
**Final Comment by Authors for AC (1)**

Dear AC,

We sincerely appreciate your efforts in managing the review process under such challenging circumstances.

For your convenience, here we provide a overview of discussions that took place during the rebuttal period. In this comment, we provide an overview of our paper, a summary of the reviewer concerns, and a detailed account of the revisions made during the rebuttal period to address all raised concerns.

**Summary of the Paper:** This paper introduces MoReFlow, an unsupervised framework for cross-morphology or homogeneous but different sized-morpoholgy motion retargeting without paired data. By combining discrete motion tokenization with conditional flow matching, we learn a stable transport map between disjoint kinematic manifolds. One of our core contributions, Multi-Sample Condition Coupling, enables robust learning from independent datasets by dynamically forming pseudo-pairs, achieving outperforming in high-fidelity, controllable (e.g., preserving local style vs. global trajectory, reversible, and chain-of-retarget) motion generation compared to existing baselines.

**Summary of Reviewer comments:** The reviewers generally acknowledged the novelty and effectiveness of our flow matching approach for motion retargeting. However, they raised following several key concerns.

- **Methodological Clarity:** Several reviewers (Khj3, 2E3m) requested a clearer definition of the condition variable $c$ and a deeper rationale for the unpaired learning strategy (Multi-Sample Condition Coupling). Also, reviewer (KJNg) recommends that this core contribution should be described in main text.
- **Architectural Justification:** Reviewer 2E3m questioned our design choice of using separate encoders/decoders for each character instead of a shared encoders/decoders across characters. And, the reviewer asked what reasons for selecting this design.
- **Experimental Rigor:** Reviewers (Khj3, KJNg) noted that the baseline comparisons were limited and that the text-only ablation study lacked readability. There were also calls for more transparency regarding dataset details and preprocessing (Khj3).
- **Scalability & Scope:** Reviewers (Khj3, 2E3m) raised points about the scalability of our approach to unseen characters or scarce data.

---

> ### Author Response · Authors · 2025-12-04
> **Final Comment by Authors for AC (2)**
>
> **Summary of Revisions and Improvements:**
>
> We also have attached a document highlighting the differences for your convenience in reviewing the changes.
>
> 1. **Clarified Multi-Sample Condition Coupling:** We have rewritten **Section 3.3** to explicitly articulate how our dynamic batch-wise coupling acts as a stochastic regularization mechanism. We explain that this "one-to-many" exposure serves as a form of data augmentation, preventing mode collapse and enabling generalization. Furthermore, we added a new illustrative diagram (**Figure 3**) and expanded **Appendix A.3** with a detailed step-by-step breakdown (including mathematical formulations for coupling matrix) to provide complete transparency on the pseudo-pairing process.
> 2. **Defined Condition Variable $c$:** We clarified the definition of the condition variable $c$, explicitly stating its dual role as an "alignment objective selector" during training (determining the feature metric) and as a geometric guide during inference. This resolves confusion regarding its physical meaning.
> 3. **Justified Architectural Choices:** Based on additional validation experiments conducted during the rebuttal (in discussion with Reviewer 2E3m), we added a rationale in **Section 3.1** justifying our use of individual VQ-VAEs. We highlighted that shared latent spaces lead to representation collapse in heterogeneous settings.
> 4. **Differentiation from Prior Generative Models:** We added a concise paragraph in **Section 2** highlighting why MoReFlow adopts conditional flow matching over traditional GANs or VAEs, emphasizing its stability, lack of mode collapse, and suitability for continuous, reversible mapping.
> 5. **New Ablation Visualizations:** We replaced the previous tabular ablation results with a new set of bar plots (**Figure 4**) in **Section 4.5**. These plots clearly visualize the impact of latent space design, window size, and batch size, offering a more intuitive comparison of performance metrics (FID, DIV, ALI, NAT).
> 6. **Expanded Baseline Analysis:** We revised the main comparison table (**Table 1**) to separately report performance on "Human" and "Non-human" benchmarks, as well as distinguishing between "Locomotion" and "Upper-body" tasks. This granular breakdown reveals new insights, such as the specific strengths of our method in non-human locomotion compared to baselines like Walk-the-Dog (WtD). We also added a clarification in **Section 4.4** justifying our selection of WtD and ACE as the most relevant unsupervised baselines.
> 7. **Dataset Details:** we added a new Appendix section (**A.5**) detailing our dataset composition and preprocessing pipeline. This includes a new bar plot visualizing the distribution of motion clips across source and target domains, demonstrating that our target dataset is intentionally broader to support diverse retargeting candidates. We also provided specific references to the tool used (MINK) for kinematic adaptation to improve reproducibility.
>
> **Summary of Responses to Each Reviewer:**
>
> Here we summarize our key responses and corresponding revisions for each reviewer. For a more detailed account of the reviewers’ concerns and our concrete, corresponding responses, we kindly refer you to the discussion section.
>
> 1. **Reviewer Khj3:** Addressed concerns about the definition of $c$ and the rationale behind Multi-Sample Condition Coupling. Clarified that our method handles disparate motion coverage (e.g., different speed limits) by prioritizing "best-effort" semantic matches rather than requiring strict distributional overlap.  Added dataset details and preprocessing steps in Appendix A.5. Expanded baseline analysis to differentiate human/non-human performance. Explained and added the justification of our choice on baslines for cross-morphology retargeting comparison.
> 2. **Reviewer 2E3m:** Conducted a new experiment comparing "Shared" vs. "Separate" architectures, demonstrating that shared models suffer from representation collapse. Please refer the discussion to see more detailed for this experiment. Clarified the Multi-Sample Condition Coupling mechanism and the role of $c$. Addressed concerns about temporal alignment by emphasizing our use of high-dimensional condition features.
> 3. **Reviewer KJNg:** Improved the presentation of ablation studies with new figures (Figure 4). Added detailed pseudocode and explanations for the coupling strategy in Appendix A.3.
> 4. **Reviewer rBVb:** Addressed scalability limitations in the discussion, clarifying the scope of our current contribution. And, we have a chance to discuss future idea for improving the scalability.
>
> We believe these revisions comprehensively address the reviewers' comments and strengthen the paper's contribution.
>
> Sincerely,
>
> Authors of MoReFlow

---

### Meta-Review · Area_Chair_ibdg · 2026-01-07

**Summary:**

This submission proposes MoReFlow, an unsupervised motion-retargeting framework that aligns character-specific motion token spaces using conditional flow matching and a Multi-Sample Condition Coupling strategy to create pseudo-pairs from unpaired datasets. Reviewers agreed the overall direction is interesting, but there are major concerns that remain: 1) limited experimental evaluation: limited baselines comparison and insufficient qualitative evidence for key claims; 2) Without sufficient evidence, it is not clear about the performance of scalability or generalization as claimed by the authors. Therefore, the final recommendation is Reject.

**Reviewer Concerns:**

Addressed: 1) Methodological clarity about Multi-Sample Condition Coupling: multiple reviewers have raised this question, and the authors provided more detailed step-by-step algorithm explanation, although it is in the Appendix; 2) separate vs shared tokenizers: The authors report additional validation and added a justification that shared latent spaces can collapse in heterogeneous settings.

Not fully addressed: 1) How to choose or set the condition variable, especially during the inference; 2) For separate tokenizers, how to make sure the latent spaces are shared between characters? 3) Reviewers flagged that baseline comparisons are limited and that the ablation evidence is hard to evaluate without stronger qualitative support.

It is actually very difficult to find out the content in the paper that was revised during the rebuttal. It is better to highlight the modification.

**Reviewer Scores:**

Reviewer Khj3 might not raise the score, given that the baseline comparison, as questioned by this reviewer, is not provided. Reviewer 2E3m had several rounds of discussion with the authors, but didn't show any intention to raise the score. Reviewer KJNg might keep the score, while Reviewer rBVb stated to increase the score.

---

### Decision · Program_Chairs · 2026-01-26

Reject